# Hierarchical Encoding Tree with Modality Mixup for Cross-modal Hashing

**Zhiping Xiao[1], Junyu Luo[2†], Hang Zhou[3], Yusheng Zhao[2], Xiao Luo[4†], Pengyun Wang[5], Wei Ju[2], Siyu Heng[6], Ming Zhang[2†]**

[1] Paul G. Allen School of Computer Science and Engineering, University of Washington
[2] State Key Laboratory for Multimedia Information Processing,
School of Computer Science, PKU-Anker LLM Lab, Peking University
[3] Department of Statistics and Operations Research & School of Data Science and Society,
University of North Carolina at Chapel Hill
[4] Department of Statistics, University of Wisconsin–Madison
[5] Data Science Institute, University of Chicago
[6] Department of Biostatistics, New York University
`patxiao@uw.edu, luo.junyu@outlook.com,`
`xiao.luo@wisc.edu, mzhang_cs@pku.edu.cn`

## Abstract

Cross-modal retrieval is a fundamental task that aims to learn semantic corre-spondences across different data modalities, such as visual and textual modalities. Unsupervised hashing methods can efficiently manage large-scale data and can be effectively applied to cross-modal retrieval studies. However, existing meth-ods typically fail to fully exploit the hierarchical semantic structure within text and image data, where instances naturally organize into multi-level communities of varying granularity. Moreover, the commonly-used direct modal alignment cannot effectively bridge the semantic gap between these two modalities. To ad-dress these issues, we introduce a novel **Hi**erarchical Encodi**n**g **T**ree with Modal-ity Mixup (**HINT**) method, which achieves effective cross-modal retrieval by ex-tracting hierarchical cross-modal relations. HINT constructs a cross-modal en-coding tree guided by hierarchical structural entropy and generates *proxy* samples of text and image modalities for each instance from the encoding tree. Through the curriculum-based mixup of proxy samples, HINT achieves progressive modal alignment and effective cross-modal retrieval. We also conduct cross-modal con-sistency learning to achieve global-view semantic alignment between text and im-age representations. Extensive experiments on a range of cross-modal retrieval datasets demonstrate the superiority of HINT over state-of-the-art methods.

## 1 Introduction

Cross-modal retrieval aims to measure the semantic similarity between different modalities, using retrieval methods such as approximate nearest neighbors (ANNs) search (Zhu et al., 2023; Zhen et al., 2019). Cross-modal retrieval has significant application value, such as in retrieval-augmented generation (RAG) (Li et al., 2024b; Cui et al., 2024) and search engines (Song et al., 2024; Chen et al., 2017). With the rapid development of large-scale vision-language datasets, cross-modal re-trieval has attracted increasing attention. Therefore, researchers have turned to hashing-based cross-modal retrieval (Cao et al., 2017; Yan et al., 2020), which achieves efficient storage and indexing by substituting bit-wise operations for computationally prohibitive pairwise distance comparisons. Hashing-based cross-modal retrieval methods map high-dimensional semantic vectors from differ-ent modalities into a unified Hamming space (binary space) (Luo et al., 2023; Huang et al., 2024), enabling similarity comparison and ANNs.

---

[†]Corresponding authors.

Cross-modal hashing has garnered significant attention from the community (Zhang et al., 2024b; Sun et al., 2024; Luo et al., 2025). It includes supervised and unsupervised methods. Supervised cross-modal hashing (Shen et al., 2024; Ma et al., 2024; Liu et al., 2019a; Lu et al., 2019) learn hash codes using labeled data. However, due to the expensiveness and scarcity of cross-modal annotations in real-world scenarios (Wang et al., 2023), researchers have shifted their focus to unsupervised methods that do not rely on annotations. Unsupervised cross-modal hashing (Liang et al., 2024; Li et al., 2024a; Zhang et al., 2018) leverage pair-wise cross-modal data, exploiting the similarity between samples and employing mechanisms such as contrastive learning (Hu et al., 2022) and adversarial learning (Li et al., 2019) to guide hash learning.

Unsupervised cross-modal hashing has made promising progress (Liang et al., 2024; Li et al., 2024a; Zhang et al., 2024b; Tu et al., 2023), but still suffers from the following issues: The first challenge is **the lack of hierarchical semantic structure**. Without semantic annotations (*e.g.*, category labels), previous unsupervised methods can only leverage paired data (Hu et al., 2022) (*e.g.*, image-text pairs) as supervision, which provides flat (*i.e.*, no hierarchical structure, all data points are on the same level) and sparse signals. However, real-world text and image data naturally exhibit a hierarchical semantic structure, containing numerous local communities. The instances within each community have similar semantics, while the semantic differences across communities are substantial. The absence of hierarchical-information-mining leads to insufficient exploration of community relationships, hindering the learning of generalizable hash codes. Furthermore, this challenge is compounded by **the ineffective alignment of heterogeneous modalities**. Existing methods (Hu et al., 2022; Zhang et al., 2024b; Tu et al., 2023) employ different encoders to project data from various modalities and optimize towards a common objective. Nevertheless, due to the inherent heterogeneity across modalities (*e.g.*, manifested in structure and semantics), direct alignment can pose a high learning difficulty and lead to suboptimal performance. Therefore, it is necessary to conduct hierarchical cross-modal learning and in a progressive manner.

To address the aforementioned issues, we propose a novel approach called **Hi**erarchical Codi**n**g **T**ree with Modality Mixup (**HINT**) for hashing-based cross-modal retrieval. The core idea of HINT lies in constructing a cross-modal encoding tree that recovers hierarchical semantic structures and mines local semantic communities. Specifically, guided by the hierarchical structural entropy (Li et al., 2018; Zou et al., 2023), we con-

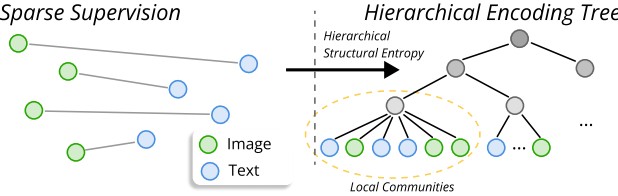

Figure 1: HINT transforms sparse cross-modal supervision (left) into a meaningful hierarchical encoding tree (right), which reveals local semantic communities for robust cross-modal alignment.

struct the encoding tree from the enhanced cross-modal relationship graph. The encoding tree has dense connections within local communities and sparse connections between communities. By utilizing the cross-modal encoding tree, we mitigate the performance degradation associated with flat and sparse cross-modal connections. Next, we synthesize proxy samples in different modalities for each sample based on the encoding tree. Through curriculum-based mixup on these proxy samples, we achieve progressive modality alignment, circumventing the challenging task of directly aligning heterogeneous modalities. We also achieve semantic alignment from a global perspective by optimizing the consistency of the semantic distributions of the proxy samples in different modalities. Extensive experiments on benchmark datasets demonstrate the superior performance of HINT.

The main contributions of this paper are: ❶ *Hierarchical Modeling Approach.* We connect the encoding tree with cross-modal hashing problems. Specifically, we construct a cross-modal encoding tree to explore the cross-modal relationships and uncover local semantic communities hierarchically. ❷ *Novel Integrated Framework.* Based on the hierarchical encoding tree, we extract cross-modal proxy samples. Leveraging the proxy samples, we design a curriculum-based modality mixup mechanism for effective cross-modal hash learning. These components synergistically form an end-to-end framework where each enables and enhances the others. On the other hand, we achieve global-view consistency learning through the distribution alignment. ❸ *Competitive Performance.* Comprehensive experiments demonstrate that HINT achieves state-of-the-art performance.

## 2 RELATED WORKS

***Cross-modal Retrieval*** is a fundamental task for bridging data of different modalities (Lee et al., 2024; Chen et al., 2024; Li et al., 2023; Krojer et al., 2022; Radford et al., 2021; Ge et al., 2024). Due to the diverse distributions and structures of texts and images, it is necessary to map them effectively into a unified representation space to calculate the semantic similarity between samples (Ding et al., 2016b; Liu et al., 2019b). With this unified representation, we can employ the approximate nearest neighbors (ANNs) (Zhu et al., 2023; Zhen et al., 2019) methods for similarity search. Recently, researchers turn to cross-modal hashing methods to enhance efficiency for both storage costs and computational costly retrieval processes (Xu et al., 2017; Jiang & Li, 2019; Wang et al., 2024b).

***Unsupervised Cross-modal Hashing*** (Liang et al., 2024; Li et al., 2024a; Zhang et al., 2024b; Wang et al., 2024b) utilizes data correlation information to project cross-modal data into a unified binary space (Huang et al., 2024; Wu et al., 2018). Due to the expensive and difficult acquisition of labeled cross-modal data, supervised methods sometimes face challenges in the real-world (Hu et al., 2022). Therefore, unsupervised cross-modal hashing has attracted widespread attention (Zhou et al., 2014; Gao et al., 2023; Mikriukov et al., 2022). Researchers also use adversarial networks (Li et al., 2019) and contrastive learning (Hu et al., 2022) to handle cross-modal hash learning. These methods generally rely on sparse text-image relationships, lacking local community mining. We explore the hierarchical cross-modal relationship and learn more generalizable hash representations through modality mixup and cross-modal consistency learning.

***Cross-modal Relationship Modeling*** is an essential topic in multi-modal research (Oh et al., 2023; Wang et al., 2024a; Li et al., 2024c; Liang et al., 2022; Huang et al., 2021). Some methods employ similarity modeling within modalities (Zhang et al., 2018), such as constructing graph structures for images and texts separately using the Wasserstein metric. The tree-based methods (Ge et al., 2021; Chen et al., 2022) are also introduced for cross-modal relationship modeling. Recent hierarchical methods (Jin et al., 2021; Zhang et al., 2025) construct predefined multi-level structures using external tools or preset granularities for supervised retrieval tasks. Other works (Sun et al., 2023) adopt architectural hierarchies through progressive projection networks for hashing.

## 3 PRELIMINARY

In this work, we consider cross-modal hash retrieval problems. The objective is to map samples from both modalities into the shared Hamming space, enabling efficient cross-modal retrieval. Specifically, let the visual vector space be $\mathcal{D}^v = \{\boldsymbol{f}_i^v\}_{i=1}^N$ and the text vector space be $\mathcal{D}^t = \{\boldsymbol{f}_i^t\}_{i=1}^N$, which is encoded by standard pre-trained visual and text encoders. We have $N$ image-text pairs without label information. We employ neural networks $\phi^v(\cdot)$ and $\phi^t(\cdot)$ to map each visual and text feature vector $\boldsymbol{f}_i^v$ and $\boldsymbol{f}_i^t$ into the Hamming space as:

$$\boldsymbol{b}_i^v = \text{sign}\left(\phi^v\left(\boldsymbol{f}_i^v\right)\right), \quad \boldsymbol{b}_i^t = \text{sign}\left(\phi^t\left(\boldsymbol{f}_i^t\right)\right), \tag{1}$$

where $\boldsymbol{b}_i^v$ and $\boldsymbol{b}_i^t$ are $L$-length hash codes, *i.e.*, $\boldsymbol{b}_i^* \in \{-1, +1\}^L$, $* \in \{v, t\}$, and $\text{sign}(\cdot)$ is the sign function. The hash codes could be used for subsequent efficient retrieval. Therefore, we need to minimize the Hamming distances between semantically similar samples across modalities while maximizing the distances between dissimilar ones. The Hamming distance is calculated as $d(\boldsymbol{b}_i^*, \boldsymbol{b}_j^*) = \frac{1}{2}(L - \langle \boldsymbol{b}_i^*, \boldsymbol{b}_j^* \rangle)$. where $L$ is the code length and $\langle \cdot, \cdot \rangle$ denotes the inner product.

## 4 METHODOLOGY

### 4.1 FRAMEWORK OVERVIEW

Sparse cross-modal connections in unsupervised scenarios pose challenges for modality alignment and cross-modal retrieval. The core idea of HINT is to establish a cross-modal encoding tree to recover the hierarchical structure across modalities and enhance the connections among local semantic clusters. Specifically, as illustrated in Figure 2, our HINT comprises three main components: ❶ *Hierarchical encoding tree construction.* Guided by the hierarchical structural entropy, we optimize the enhanced cross-modal relationship graph to obtain the encoding tree. ❷ *Cross-modal hash learning with modality mixup.* To minimize the heterogeneous gap between two modalities, HINT builds

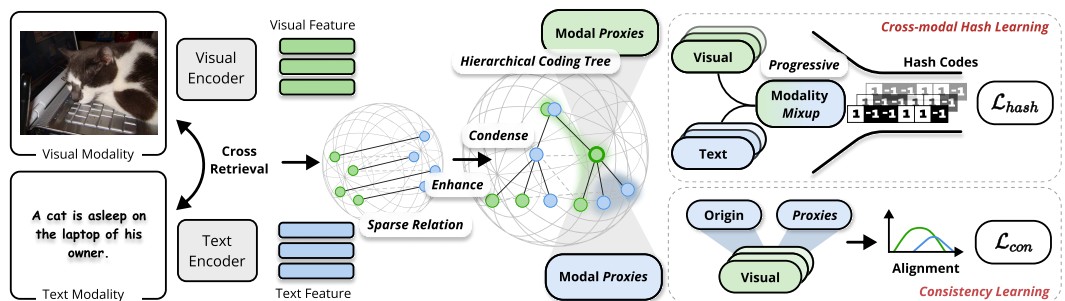

Figure 2: Overview of HINT, which constructs a hierarchical encoding tree from sparse cross-modal relationships. It then synthesizes modality proxies and performs progressive modality mixup and global-view consistency learning.

proxy samples for different modalities and progressively aligns them through a curriculum-based modality mixup mechanism (gradually transitioning from intra-modal to cross-modal alignment). ❸ *Proxy-based consistency learning.* We optimize the distribution of cross-modal proxy samples, achieving a global-level alignment.

## 4.2 HIERARCHICAL CODING TREE CONSTRUCTION

To address the challenges posed by flat and sparse cross-modal connections, we construct a hierarchical encoding tree in an *enhance*-and-*condense* manner, as shown in Figure 3. First, we enhance the intra-modal connections within the relation graph. Then, guided by the structure entropy (Li et al., 2018; Zou et al., 2023), we condense the relation graph to obtain the hierarchical encoding tree. The encoding tree exhibits a hierarchical community structure, facilitating the hash learning.

**Enhance.** In unsupervised cross-modal retrieval, we primarily rely on cross-modal pairwise relations. We first construct a inter-modal relation graph $\mathcal{G}_{inter} = \{\mathcal{V}, \mathcal{E}_{inter}\}$, where $\mathcal{V} = \mathcal{D}^v \cup \mathcal{D}^t$ and $\mathcal{E}_{inter} = \{\boldsymbol{f}_i^v, \boldsymbol{f}_i^t\}_{i=1}^N$. Since cross-modal pairs only provide sparse supervision signals in unsupervised scenarios, we strategically employ KNN to enhance intra-modal relationships. This enables us to capture fine-grained local similarities and form cohesive bottom-level semantic communities, which serve as a robust foundation for subsequent hierarchical modeling. Since $\mathcal{G}_{inter}$ is sparse and inadequate for cross-modal learning, we enrich the intra-modal relationships to construct tightly-knit low-level communities. Specifically, we turn to the cosine similarity within modality by $S_{(i,j)}^* = \cos(\boldsymbol{f}_i^*, \boldsymbol{f}_j^*)$, $* \in \{v, t\}$. We choose cosine similarity as it focuses on semantic directional alignment by normalizing vector magnitudes, which is crucial for cross-modal feature comparison. We then construct the intra-modal relationship graph $\mathcal{G}_{intra} = \{\mathcal{V}, \mathcal{E}_{intra}\}$ based on the similarity matrix by KNN manner:

$$\mathcal{E}_{intra} = \left\{ \left\{ \boldsymbol{f}_i^*, \boldsymbol{f}_j^* \right\} \mid j \in \text{topk}\left(\boldsymbol{f}_i^*, S, k\right) \right\}_{i=1}^N , \tag{2}$$

where $k$ is set to 3 according to the hyperparameter study in Section 5.2. This choice balances information gain and noise robustness: larger $k$ introduces noisy relationships, while smaller $k$ fails to capture sufficient local semantics. Then, we merge the intra-modal relationships $\mathcal{G}_{intra}$ and the inter-modal relationships $\mathcal{G}_{inter}$ to obtain the cross-modal relationship graph $\mathcal{G}_{cross} = \{\mathcal{V}, \mathcal{E}_{intra} \cup \mathcal{E}_{inter}\}$.

**Condense.** To simplify the cross-modal semantics and construct the hierarchical relationship, we extract the encoding tree $\mathcal{T}$ from $\mathcal{G}_{cross}$ with the guidance of structural entropy, as illustrated in the center part of Figure 2. We first introduce the 1-D structural entropy as:

$$E^1(\mathcal{G}) = -\sum_{\boldsymbol{v} \in V} \frac{\boldsymbol{d}_v}{vol(\mathcal{G})} \log \frac{\boldsymbol{d}_v}{vol(\mathcal{G})} , \tag{3}$$

where $\boldsymbol{d}_v$ denotes the degree for node $v$ and $vol(\cdot)$ denotes the sum of degree for nodes in $\mathcal{G}$. Then, we introduce the hierarchical structural entropy for the $\mathcal{T}$ as:

$$E^{\mathcal{T}}(\mathcal{G}) = -\sum_{\alpha \in \mathcal{T}} \underbrace{\frac{g_\alpha}{vol(\mathcal{G})}}_{\substack{\text{information} \\ \text{leakage}}} \log \underbrace{\frac{\mathcal{V}_\alpha}{\mathcal{V}_{\alpha^-}}}_{\substack{\text{encoding} \\ \text{efficiency}}} . \tag{4}$$

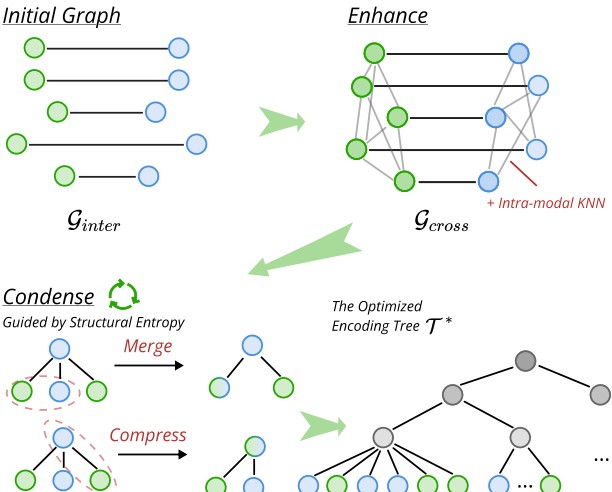

Figure 3: The construction pipeline of the encoding tree, involving graph enhancement and entropy-guided condensing operations.

Intuitively, the factor $\frac{g_\alpha}{vol(\mathcal{G})}$ captures the information leakage of community $\alpha$, while the logarithmic term $\log \frac{\mathcal{V}_\alpha}{\mathcal{V}_{\alpha^-}}$ reflects the encoding efficiency of the hierarchy. In Eq. 4, $\mathcal{G}$ is $\mathcal{G}_{cross}$, $\alpha$ is a node in $\mathcal{T}$, $\mathcal{T}_\alpha$ is the subtree rooted at $\alpha$, $\mathcal{T}_{\alpha^-}$ is the subtree rooted at $\alpha$'s parent node, $g_\alpha$ is the number of intra-modal relation links originating from the subtree $\mathcal{T}_\alpha$, $\mathcal{V}_\alpha$ is the sum of degrees in subtree $\mathcal{T}_\alpha$, and $\mathcal{V}_{\alpha^-}$ is the sum of degrees in subtree $\mathcal{T}_{\alpha^-}$. Eq. 4 generalizes Eq. 3 to hierarchical communities and reduces to Eq. 3 when the hierarchy collapses to a flat partition.

Guided by structural entropy, we convert the cross-modal relation graph into an encoding tree through *Merge* and *Compress*, as shown in Figure 3. Firstly, we perform node *Merge* operation to generate a binary encoding tree. The node *Merge* operation merges nodes that belong to the same parent node. These nodes may be highly semantically similar, and merging them can reduce the overall structural entropy of the cross-modal encoding tree. The node *Merge* operation is defined as: $\mathcal{T}' = Merge_{\mathcal{T}}(\alpha, \beta)$, we check all nodes if it can introduce decrease in $E^{\mathcal{T}}(\mathcal{G})$.

Secondly, *Compress* operation is performed to optimize the encoding tree, mainly targeting adjacent nodes at different levels, and constructing local clusters. This is achieved by attempting to move the child encoding tree with $\alpha$ as the root to its parent node's parent node, thereby enabling compression of the cross-modal semantic graph. After encoding tree compression, if there are no child encoding trees connected to a parent node, this parent node can be contracted to its parent node. The *Compress* operation is defined as: $\mathcal{T}' = Compress_{\mathcal{T}}(\alpha, \beta)$. Similarly, we check the nodes and conduct the *Compress* operation if it can introduce entropy decrease. Overall, we optimize the cross-modal encoding tree following a greedy principle. Specifically, we traverse the tree nodes in a breadth-first search manner. We attempt the aforementioned operations, and if they can decrease the structural entropy, we execute the operation. While this greedy approach does not guarantee global optimality, it provides an efficient and practical solution that converges to stable structures with clear hierarchical communities, as validated by our experiments. The cross-modal encoding tree after optimization is defined as $\mathcal{T}^* = \arg\min(E^{\mathcal{T}}(\mathcal{G}))$, where $\mathcal{T}^*$ is the optimized encoding tree. The resulting tree naturally captures semantic granularity transitions, with upper nodes representing broad categories, mid-level nodes capturing fine-grained concepts, and leaf nodes corresponding to specific instances. $\mathcal{T}^*$ exhibits better cross-modal semantic properties. Specifically, they encompass more comprehensive local semantic motifs while mitigating the connections within high-density communities. These characteristics facilitate subsequent discriminative hash code learning and consistency learning.

### 4.3 STRUCTURE-GUIDED CROSS-MODAL HASH LEARNING

After obtaining the optimized cross-modal encoding tree $\mathcal{T}^*$, we use it for unsupervised hash learning. Compared to existing unsupervised cross-modal hashing works (Hu et al., 2022; Liu et al., 2017; Zhang et al., 2018; 2024b; Tu et al., 2023), our method can exploit local semantic communities, avoiding the bias caused by individual samples. Simultaneously, we jointly model both image and text modalities, mapping the vectors from distinct modalities into a unified binary space.

***Proxy Construction.*** For each sample $\boldsymbol{f}_i^*$, we sample its neighboring nodes $\mathcal{N}^+(\boldsymbol{f}_i^*)$ with the same modality and $\mathcal{N}^-(\boldsymbol{f}_i^*)$ with the opposite modality on the cross-modal encoding tree $\mathcal{T}^*$, and obtain the *proxy* samples via:

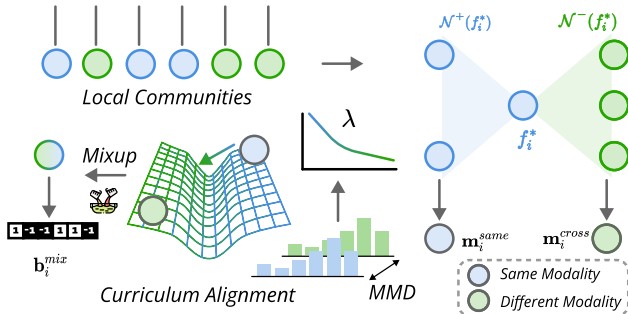

$$\boldsymbol{m}_i^{same} = \frac{1}{|\mathcal{N}^+(\boldsymbol{f}_i^*)|} \sum_{j \in \mathcal{N}^+(\boldsymbol{f}_i^*)} \phi^*\left(\boldsymbol{f}_j^*\right),$$

$$\boldsymbol{m}_i^{cross} = \frac{1}{|\mathcal{N}^-(\boldsymbol{f}_i^*)|} \sum_{j \in \mathcal{N}^-(\boldsymbol{f}_i^*)} \phi^*\left(\boldsymbol{f}_j^*\right),$$

(5)

Figure 4: The Modality Mixup pipeline: generating modality-specific proxies, using their MMD to guide a curriculum-based mixup, and producing $\boldsymbol{b}_i^{mix}$.

where $\boldsymbol{f}_i^*, * \in \{v, t\}$ is the vector, $\phi^*(\cdot)$ is the hash model we introduced in Eq. 1. The neighbor counts $|\mathcal{N}^+(\boldsymbol{f}_i^*)|$ and $|\mathcal{N}^-(\boldsymbol{f}_i^*)|$ are naturally determined by the tree structure $\mathcal{T}^*$ and vary across nodes. Therefore, $\boldsymbol{m}_i^{same}$ for a text sample $\boldsymbol{f}_i^t$ (aggregating text neighbors) is distinct from $\boldsymbol{m}_i^{same}$ for a visual sample $\boldsymbol{f}_i^v$ (aggregating visual neighbors), as they are derived from different sets of modality-specific neighbors. A similar distinction applies to $\boldsymbol{m}_i^{cross}$. The modal *proxy* samples can be viewed as a mediator between the two modalities, consisting of semantically similar nodes from the opposite modality, exhibiting better semantic robustness.

***Modality Mixup.*** Leveraging *proxy* samples, we introduce a mixup mechanism (Oh et al., 2023; Verma et al., 2019; Huang et al., 2022) for progressively learning. Specifically, as Figure 4, $\boldsymbol{m}_i^{same}$ and $\boldsymbol{m}_i^{cross}$ are mixup to generate the hash codes:

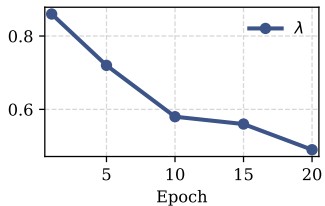

$$\boldsymbol{b}_i^{mix} = \text{sign}(\frac{\lambda}{1+\lambda}\boldsymbol{m}_i^{same} + \frac{1}{1+\lambda}\boldsymbol{m}_i^{cross}),$$

$$\lambda = \widehat{\text{MMD}}\left(\rho\left(\boldsymbol{m}^{same}, \mathcal{B}\right), \rho\left(\boldsymbol{m}^{cross}, \mathcal{B}\right)\right),$$

(6)

where $\lambda$ is to measure the distribution difference of different modalities, and $\rho$ is the cosine distance metrics, we sample in mini-batch $\mathcal{B}$ and calculate $\widehat{\text{MMD}}$ using the Gaussian ker-

Figure 5: Evolution of $\lambda$ during learning on MIRFlickr-25K.

nel (Long et al., 2015; Tolstikhin et al., 2016). Notably, $\lambda$ is not a manually tuned hyperparameter but a data-driven metric computed via MMD, which dynamically quantifies the modality gap during training. This adaptive weighting mechanism implements a curriculum learning strategy (Hu et al., 2025). The theoretical foundation lies in manifold regularization (Verma et al., 2019): by interpolating in feature space between semantically purified proxy samples, we encourage smooth decision boundaries and force both modalities to find a shared semantic space where their interpolation remains meaningful for hash learning. As Figure 5, the evolution of $\lambda$ empirically demonstrates this progressive alignment process.

***Cross-modal Hash Learning.*** We employ the mixed hash code $\boldsymbol{b}_i^{mix}$ for hash learning. In the learning process, we sample in batches, and the other samples in the batch can serve as negative samples, enhancing the discriminative power of the hash codes (Oh et al., 2023; Luo et al., 2020). The objective function for cross-modal hash learning is:

$$\mathcal{L}_{hash} = -\sum_{i=1}^{N}\left(\log\frac{\exp\left(\langle\boldsymbol{f}_i^*, \boldsymbol{b}_i^{mix}\rangle/\tau\right)}{\sum_{j=1}^{|\mathcal{B}|}\exp\left(\langle\boldsymbol{f}_i^*, \boldsymbol{b}_j^{mix}\rangle/\tau\right)}\right),$$

(7)

where $* \in \{v, t\}$ and $\tau$ is the temperature parameter, which is set to $0.3$ according to Section 5.2.

In HINT, we utilize the cross-modal encoding tree to guide hash learning. Since our cross-modal encoding tree has more comprehensive connections on local base groups, it can help align hash codes to more robust semantics. Meanwhile, the cross-modal encoding tree is sparser between communities, facilitating discriminability between groups and achieving discriminative hash codes.

***Theoretical Discussion.*** The main benefit of HINT comes from the optimized encoding tree $\mathcal{T}^*$, which converts sparse pair-wise supervision (Oh et al., 2023) into local semantic communities. Under a local community consistency assumption, the same-modal and cross-modal proxies are unbiased estimators of the underlying community prototype, and their weighted mixture $\tilde{z}_i$ has variance

$O(\alpha_i^2/|\mathcal{N}_i^+(f_i^*)| + (1-\alpha_i)^2/|\mathcal{N}_i^-(f_i^*)|)$. Therefore, the mixed proxy becomes a low-variance semantic target whose expected distance to the anchor is smaller than that of proxies from other communities whenever the inter-community separation dominates the proxy variance. This provides a theoretical explanation for why HINT can improve retrieval under the local community consistency model by turning sparse sample-level alignment into margin-preserving community-level alignment.

**Lemma 4.1.** *Let $z_i^* = \tanh(\phi_*(f_i^*))$ be the continuous training-time code, where $* \in \{v, t\}$, and let $\bar{*}$ denote the opposite modality. Define*

$$m_i^{\mathrm{same}} = \frac{1}{|\mathcal{N}_i^+(f_i^*)|} \sum_{j \in \mathcal{N}_i^+(f_i^*)} z_j^*, \quad m_i^{\mathrm{cross}} = \frac{1}{|\mathcal{N}_i^-(f_i^*)|} \sum_{j \in \mathcal{N}_i^-(f_i^*)} z_j^{\bar{*}},$$

*$\tilde{z}_i = \alpha_i m_i^{\mathrm{same}} + (1-\alpha_i)m_i^{\mathrm{cross}}$ and $\alpha_i = \lambda_i/(1+\lambda_i)$. Assume that the neighbors selected by the optimized encoding tree $T^*$ for sample $i$ belong to a latent semantic community $c(i)$ with prototype $\mu_{c(i)}$, and that each neighbor code can be written as $z_j = \mu_{c(i)} + \varepsilon_j$, $\mathbb{E}[\varepsilon_j] = 0$, $\mathbb{E}\|\varepsilon_j\|_2^2 \le \sigma^2$, with mutually independent noises. Then*

$$\mathbb{E}\|\tilde{z}_i - \mu_{c(i)}\|_2^2 \le \sigma^2 \left( \frac{\alpha_i^2}{|\mathcal{N}_i^+(f_i^*)|} + \frac{(1-\alpha_i)^2}{|\mathcal{N}_i^-(f_i^*)|} \right) =: r_i^2. \tag{8}$$

*For sample $k$, define $\tilde{z}_k$ analogously and assume it satisfies the same tree-induced proxy model around $\mu_{c(k)}$, so that $\tilde{z}_k = \mu_{c(k)} + \eta_k$ with $\mathbb{E}[\eta_k] = 0$. If in addition the anchor code satisfies $z_i^* = \mu_{c(i)} + \xi_i$, where $\mathbb{E}[\xi_i] = 0$ and $\xi_i$ is independent of the proxy noises, then for any sample $k$ from a different community $c(k) \neq c(i)$,*

$$\mathbb{E}\|z_i^* - \tilde{z}_k\|_2^2 - \mathbb{E}\|z_i^* - \tilde{z}_i\|_2^2 \ge \|\mu_{c(i)} - \mu_{c(k)}\|_2^2 - r_i^2. \tag{9}$$

Equation 9 shows that whenever the inter-community separation $\|\mu_{c(i)} - \mu_{c(k)}\|_2^2$ exceeds the proxy variance term $r_i^2$, the tree-guided mixed proxy is expected to be closer to its anchor than proxies from other communities, yielding a positive retrieval margin.

### 4.4 PROXY-BASED CONSISTENCY LEARNING

Due to the heterogeneous gap between modalities, it is necessary to introduce an additional global-view modal alignment mechanism to achieve better alignment and enhance the generalization ability.

***Semantic Consistency Learning***. We leverage the modal *proxy* samples for modality alignment learning, assuming that the original samples and their *proxy* samples should have similar semantic positions and similar distribution. Specifically, the modality semantics in a global view can be represented as the similarity distribution between samples and other samples within the same batch:

$$p(f_i^*) = \left[ \rho\left(f_i^*, f_j^*\right) \mid f_j^* \in \mathcal{B}^- \right], \tag{10}$$

where $\mathcal{B}^-$ includes the opposite modality instances within the same mini-batch, and $\rho(\cdot)$ is the cosine similarity function. Our objective is achieved by optimizing the KL divergence:

$$\mathcal{L}_{con} = \sum_{i=1}^{|\mathcal{B}|} \left( D_{KL}\left(p\left(f_i^*\right) \parallel p\left(m_i^{cross}\right)\right) \right), \tag{11}$$

where $|\mathcal{B}|$ is the batch size, $p\left(f_i^*\right)$ and $p\left(m_i^{cross}\right)$ are the semantic distributions of the $i$-th sample and its cross-modal *proxy* sample, respectively. By optimizing the consistency learning $\mathcal{L}_{con}$, we achieve modality alignment learning at a high level by leveraging the semantic-stable *proxy* samples.

***Summary.*** Our method constructs a hierarchical encoding tree by unsupervised cross-modal mining and simultaneously leverages the encoding tree for cross-modal hash learning and semantic consistency learning. During the testing phase, the hierarchical encoding tree and proxy samples are not used. We directly employ the corresponding hash model to generate its hash code. This design ensures efficient retrieval with minimal computational overhead during inference time. Due to the non-differentiability of the $\mathrm{sign}(\cdot)$ function, it is challenging to optimize the overall objective. Therefore, we adopt $\tanh(\cdot)$ as a surrogate during the training process. The whole algorithm is summarized in Algorithm 1 and Appendix A. The computational complexity and time efficiency are discussed in the Appendix C.

Table 1: Comparison of MAP performance (%) across various cross-modal hashing methods.

| Methods | MIRFlickr-25K | | | | NUS-WIDE | | | | MS-COCO | | | |
|---|---|---|---|---|---|---|---|---|---|---|---|---|
| | 16 | 32 | 64 | 128 | 16 | 32 | 64 | 128 | 16 | 32 | 64 | 128 |
| *Image → Text* | | | | | | | | | | | | |
| CVH | 62.0 | 60.8 | 59.4 | 58.3 | 48.7 | 49.5 | 45.6 | 41.9 | 50.3 | 50.4 | 47.1 | 42.5 |
| LSSH | 59.7 | 60.9 | 60.6 | 60.5 | 44.2 | 45.7 | 45.0 | 45.1 | 48.4 | 52.5 | 54.2 | 55.1 |
| CMFH | 55.7 | 55.7 | 55.6 | 55.7 | 33.9 | 33.8 | 34.3 | 33.9 | 36.6 | 36.9 | 37.0 | 36.5 |
| FSH | 58.1 | 61.2 | 63.5 | 66.2 | 55.7 | 56.5 | 59.8 | 63.5 | 53.9 | 54.9 | 57.6 | 58.7 |
| MTFH | 50.7 | 51.2 | 55.8 | 55.4 | 29.7 | 29.7 | 27.2 | 32.8 | 39.9 | 29.3 | 29.5 | 39.5 |
| FOMH | 57.5 | 64.0 | 69.1 | 65.9 | 30.5 | 30.5 | 30.6 | 31.4 | 37.8 | 51.4 | 57.1 | 60.1 |
| DCH | 59.6 | 60.2 | 62.6 | 63.6 | 39.2 | 42.2 | 43.0 | 43.6 | 42.2 | 42.0 | 44.6 | 46.8 |
| DGCPN | 65.1 | 68.3 | 71.8 | 72.4 | 60.1 | 61.8 | 63.1 | 64.0 | 55.6 | 56.9 | 57.8 | 58.0 |
| UCHSTM | 70.1 | 71.5 | 72.4 | 72.3 | 62.5 | 63.5 | 64.6 | 64.4 | 55.8 | 57.2 | 57.6 | 57.3 |
| UCCH | 71.6 | 72.6 | 72.8 | 73.2 | 62.1 | 62.3 | 64.0 | 64.5 | 56.0 | 56.2 | 56.6 | 57.4 |
| UDDH | 71.4 | 72.9 | 74.0 | 74.6 | 63.7 | 64.2 | 65.1 | 65.9 | 56.8 | 57.8 | 59.0 | 59.9 |
| HuggingHash+ | 71.6 | 73.2 | 74.3 | 74.5 | 63.9 | 64.8 | 65.6 | 66.4 | 57.1 | 58.3 | 59.4 | 60.5 |
| DEMO | 71.8 | 73.3 | 73.4 | 74.3 | 64.6 | 64.8 | 66.2 | 66.4 | 57.5 | 57.8 | 58.6 | 60.5 |
| GCRH | 71.0 | 72.2 | 72.7 | 73.3 | 63.9 | 64.0 | 65.3 | 65.5 | 56.8 | 57.0 | 57.8 | 59.6 |
| VTM-UCH | 71.9 | 73.6 | 73.9 | 74.5 | 64.6 | 65.1 | 66.0 | 66.6 | 57.6 | 58.2 | 58.8 | 60.3 |
| **HINT** | **72.9** | **74.4** | **75.1** | **75.5** | **65.1** | **65.5** | **66.5** | **67.3** | **58.5** | **59.5** | **60.4** | **61.1** |
| *Text → Image* | | | | | | | | | | | | |
| CVH | 62.9 | 61.5 | 59.9 | 58.7 | 47.0 | 47.5 | 44.4 | 41.2 | 50.6 | 50.8 | 48.6 | 42.9 |
| LSSH | 60.2 | 59.8 | 59.8 | 59.7 | 47.3 | 48.2 | 47.1 | 45.7 | 49.0 | 52.2 | 54.7 | 56.0 |
| CMFH | 55.3 | 55.3 | 55.3 | 55.3 | 30.6 | 30.6 | 30.6 | 30.6 | 34.6 | 34.6 | 34.6 | 34.6 |
| FSH | 57.6 | 60.7 | 63.5 | 66.0 | 56.9 | 60.4 | 65.1 | 66.6 | 53.7 | 52.4 | 56.4 | 57.3 |
| MTFH | 51.4 | 52.4 | 51.8 | 58.1 | 35.3 | 31.4 | 39.9 | 41.0 | 33.5 | 37.4 | 30.0 | 33.4 |
| FOMH | 58.5 | 64.8 | 71.9 | 68.8 | 30.2 | 30.4 | 30.0 | 30.6 | 36.8 | 48.4 | 55.9 | 59.5 |
| DCH | 61.2 | 62.3 | 65.3 | 66.5 | 37.9 | 43.2 | 44.4 | 45.9 | 42.1 | 42.8 | 45.4 | 47.1 |
| DGCPN | 65.3 | 68.2 | 71.2 | 71.5 | 60.5 | 62.6 | 63.7 | 64.4 | 55.0 | 56.6 | 57.8 | 57.7 |
| UCHSTM | 69.5 | 71.1 | 71.3 | 72.3 | 63.2 | 64.3 | 65.1 | 65.2 | 55.5 | 56.7 | 57.8 | 57.3 |
| UCCH | 70.3 | 71.2 | 72.0 | 72.1 | 62.5 | 63.7 | 65.0 | 65.2 | 56.4 | 57.3 | 57.2 | 58.1 |
| UDDH | 70.5 | 71.6 | 72.8 | 73.5 | 64.5 | 65.2 | 66.0 | 66.6 | 56.6 | 57.5 | 58.5 | 59.4 |
| HuggingHash+ | 70.7 | 72.0 | 73.2 | 73.8 | 64.8 | 65.7 | 66.5 | 67.0 | 56.9 | 57.9 | 59.0 | 60.1 |
| DEMO | 70.8 | 71.9 | 72.2 | 72.8 | 65.4 | 65.5 | 66.9 | 67.1 | 57.2 | 57.9 | 58.3 | 59.7 |
| GCRH | 70.1 | 71.0 | 71.3 | 71.8 | 64.5 | 64.7 | 66.0 | 66.2 | 56.5 | 57.3 | 57.4 | 58.8 |
| VTM-UCH | 71.4 | 72.1 | 72.4 | 73.0 | 65.6 | 65.7 | 66.9 | 67.3 | 57.4 | 58.0 | 58.3 | 59.6 |
| **HINT** | **72.0** | **73.1** | **74.0** | **74.6** | **66.0** | **66.6** | **67.3** | **67.8** | **58.2** | **59.0** | **59.8** | **60.8** |

Table 2: Ablation studies. The component columns "KNN, Tree, Curr, Con" respectively denote intra-modal KNN, hierarchical encoding tree, curriculum-based mixup, and proxy-based learning.

| Methods | Components | | | | MIRF-25K | | NUS-WIDE | | MS-COCO | |
|---|---|---|---|---|---|---|---|---|---|---|
| | KNN | Tree | Curr | Con | I→T | T→I | I→T | T→I | I→T | T→I |
| HINT *V1* | | | | | 73.2 | 72.0 | 64.0 | 65.1 | 57.9 | 58.5 |
| HINT *V2* | ✓ | | | | 73.8 | 72.8 | 65.2 | 65.7 | 59.1 | 58.9 |
| HINT *V3* | ✓ | ✓ | | | 74.2 | 73.6 | 65.9 | 66.4 | 60.0 | 59.5 |
| HINT *V4* | ✓ | ✓ | ✓ | | 75.1 | 74.1 | 67.0 | 67.3 | 60.7 | 60.2 |
| **HINT** | ✓ | ✓ | ✓ | ✓ | **75.5** | **74.6** | **67.3** | **67.8** | **61.1** | **60.8** |

# 5 EXPERIMENT

## 5.1 EXPERIMENTAL SETTINGS

❶ **Datasets.** In order to comprehensively evaluate the performance of HINT, we conduct experiments on three popular public datasets: MIRFlickr-25K (Huiskes & Lew, 2008), NUS-WIDE (Rasiwasia et al., 2010), and MS-COCO (Lin et al., 2014). The detailed information is available in Appendix E.2. ❷ **Baselines.** We compare our method HINT with 15 baselines from related fields, including the latest state-of-the-art works. Details in Appendix E.1. ❸ **Implementation Details.** To ensure a fair comparison, we implement our method based on the latest SOTA works (Zhang et al., 2024b; Hu et al., 2022). Details in Appendix E.3.

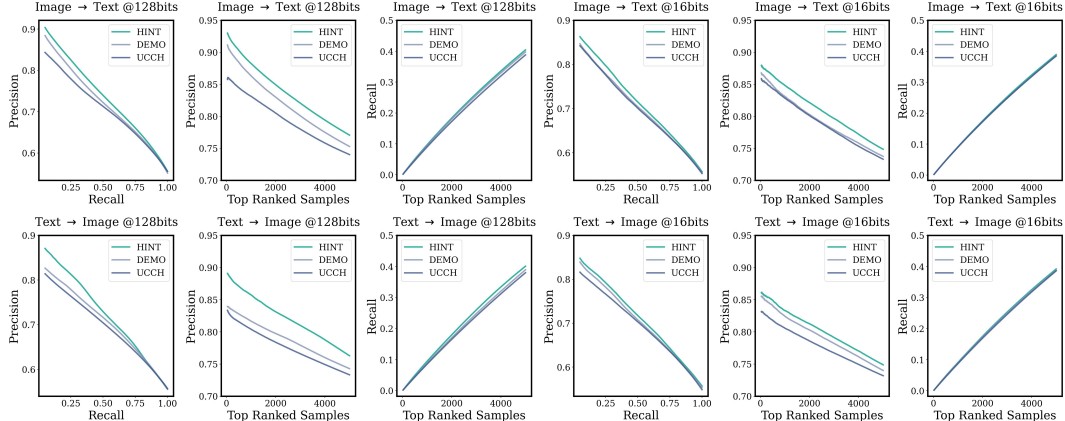

Figure 6: Hash lookup performance with 128-bit and 16-bit codes on the MIRFlickr-25K dataset. The Precision-Recall curves, Precision-N curves, and Recall-N curves are shown from left to right.

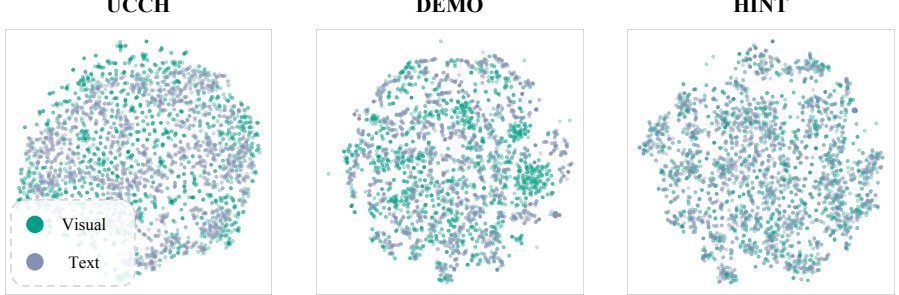

Figure 7: The t-SNE projection of hash codes from different modalities. Among competing methods, HINT shows the best ability of modal alignment.

## 5.2 RESULTS

***Hamming Ranking.*** The experiments on cross-modal retrieval benchmarks show that the proposed HINT consistently outperforms baseline approaches across different code lengths (16-128 bits). Key findings show that: ❶ Deep unsupervised hashing approaches typically outperform traditional ones, ❷ Supervised methods struggle when labeled data is limited, ❸ The method shows improved performance on challenging sub-tasks like Text→Image retrieval, and ❹ The hierarchical modeling approach proves more effective than other deep cross-modal methods for achieving progressive cross-modal alignment. Additionally, HINT demonstrates strong robustness against noisy data, maintaining superior performance even with 10% corrupted pairs. Detailed noise robustness analysis and results are provided in Appendix C.6. As shown in Table 1, performance generally improves with increasing bit length as longer hash codes provide larger Hamming space for encoding more information, though with diminishing returns at higher lengths. The more pronounced improvements on Text→Image tasks can be attributed to text feature sparsity and lower initial quality compared to visual features. HINT's hierarchical structure and proxy-based neighborhood aggregation provide more substantial benefits for the inherently more challenging tasks.

***Hash Lookup.*** To comprehensively analyze HINT's performance, we evaluate Precision-Recall, Precision-N, and Recall-N curves with 128-bit and 64-bit codes on MirFlickr-25K. From the results in Figure 6, HINT performs better in comparison to these baselines across all metrics, aligning with the MAP scores from Hamming ranking. In summary, HINT exhibited optimal performance in cross-modal hash retrieval. Experiments for other code lengths are in Appendix C.1.

***Visualization.*** We provide a t-SNE visualization analysis of HINT's performance using 128-bit hash codes on the MirFlickr-25K dataset, comparing with DEMO and UCCH, distinguishing different modalities with distinct colors. As shown in Figure 7, HINT demonstrates a superior ability to map representations from different modalities into a unified hash space, exhibiting higher alignment between text and visual modalities. The visualization suggests that HINT effectively aligns modalities and learns hash codes with generalization capabilities.

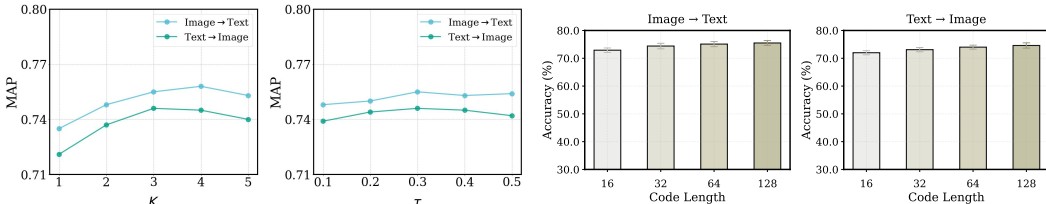

(a) Sensitive analysis of $K$ and $\tau$. HINT demonstrates robustness to hyperparameters.

(b) Stability analysis of HINT. Error bars indicate standard deviation.

Figure 8: Sensitivity and stability analyses on the MIRFlickr-25K dataset.

***Ablation Study.*** We compare the following variants of HINT: *V1,* which only uses text-image pairs without consistency learning ($\mathcal{L}_{con}$), where $\boldsymbol{b}^*_{mix}$ is obtained solely from the opposite modality; *V2,* which uses both text-image pairs and intra-modal KNN without $\mathcal{L}_{con}$, where $\boldsymbol{b}^*_{mix}$ is obtained by averaging opposite modality and KNN samples; *V3,* which constructs the hierarchical encoding tree with sample selection but without curriculum-based progressive alignment and without consistency loss $L_{con}$; *V4,* which uses $\mathcal{L}_{hash}$ consistent with the full model but excludes $\mathcal{L}_{con}$. As shown in Table 2, the full model achieves optimal performance, with hierarchical encoding tree and progressive alignment (*V3* and *V4*) yielding the most improvements. We also explored iterative tree updates but found static construction provides better efficiency-performance trade-off, with detailed analysis in Appendix C.8. Additional experiments comparing different similarity metrics for tree construction demonstrate cosine similarity's superiority over L1/L2 distances, with detailed analysis in Appendix C.7. Additional ablation studies are available in Appendix C.2.

***Sensitivity Analysis.*** We analyze the hyperparameters $K$ and $\tau$. As shown in Figure 8a, increasing $K$ from 1 to 3 improves performance on both retrieval tasks, validating the benefits of enhanced cross-modal relationships. However, further increasing $K$ to 5 introduces noisy relationships and decreases performance. Our experiments demonstrate HINT's remarkable stability, with performance fluctuation remaining within a narrow 2% margin when varying $\tau$ from 0.1 to 0.5. The model consistently outperforms baselines across most parameter settings. Based on these findings, we set $K = 3$ and $\tau = 0.3$ as the default values in our experiments.

***Stability Analysis.*** We conducted 5 independent runs with different seeds. As shown in Figure 8b, the results show that HINT exhibits remarkable stability, with performance variations consistently remaining below 1% standard deviation across different code lengths. Details in Appendix C.9.

***Time Efficiency.*** We provide a comprehensive efficiency analysis covering model complexity, training cost, and retrieval speed. As shown in Table 5, HINT achieves superior MAP performance with training efficiency comparable to DEMO and UDDH, demonstrating an excellent efficiency-performance trade-off. For retrieval efficiency, HINT and all compared hashing methods use identical Hamming distance computation on binary codes, thus sharing the same retrieval speed. Table 4 demonstrates that hash-based retrieval achieves 26-31× speedup over dense vector approaches. Details in Appendix C.5 and C.4.

## 6 CONCLUSION

This paper investigates the problem of efficient cross-modal retrieval through cross-modal hashing. We propose HINT that leverages hierarchical structural entropy to guide the construction of a cross-modal encoding tree, which has tightly connected local clusters. By incorporating progressive mixup for proxy-based alignment and consistency learning from a global perspective, we enhance the generalization capability of the produced hash codes. In the end, we demonstrate the effectiveness of our HINT via comprehensive experiments on benchmark datasets.

## ETHICS STATEMENT

Our research adheres to the ICLR Code of Ethics. All datasets used in this study are publicly available. The code and related materials will be appropriately released to ensure transparency and reproducibility of our work.

ACKNOWLEDGMENT

Ming Zhang and Junyu Luo are supported by grants from the National Natural Science Foundation of China (NSFC Grant Number 62276002).

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

# A   ALGORITHM

We present the optimization algorithm of our method in Algorithm 1. The algorithm first constructs a cross-modal relationship graph and optimizes the hierarchical encoding tree. Then during training, it performs modality mixup and consistency learning to generate effective hash codes through back-propagation.

---

**Algorithm 1** Optimization Algorithm of HINT

---

**Require**: Visual modality $\mathcal{D}^v$; Text modality $\mathcal{D}^t$;
**Ensure**: Hashing model $\phi^v(\cdot)$ and $\phi^t(\cdot)$;

1: Construct the cross-modal relationship graph $\mathcal{G}_{cross}$;
2: Optimize the hierarchical encoding tree $\mathcal{T}^*$;
3: **for** each epoch **do**
4:     **for** each batch **do**
5:         Sample $\mathcal{B}^v, \mathcal{B}^t$ from $\mathcal{D}^v, \mathcal{D}^t$;
6:         Construct proxy samples $\boldsymbol{m}^{same}, \boldsymbol{m}^{cross}$ using Eq. 5;
7:         Perform modality mixup and generate hash code $\boldsymbol{b}^{mix}$ with $\mathcal{T}^*$ by Eq. 6;
8:         Calculate the $\mathcal{L}_{hash}$ and $\mathcal{L}_{con}$;
9:         Update parameters through back-propagation;
10:     **end for**
11: **end for**

---

# B   PROOF OF LEMMA 4.1

*Proof of Lemma 4.1.* For brevity, let $n_i^+ := |\mathcal{N}_i^+(f_i^*)|$, $n_i^- := |\mathcal{N}_i^-(f_i^*)|$, $\mu_i := \mu_{c(i)}$. By the assumption and definition on the tree-selected neighbors, we write $z_j^* = \mu_i + \varepsilon_j^+$, $j \in \mathcal{N}_i^+(f_i^*)$, and $z_j^{\bar{*}} = \mu_i + \varepsilon_j^-$, $j \in \mathcal{N}_i^-(f_i^*)$, where all noises are mutually independent, satisfy $\mathbb{E}[\varepsilon_j^+] = \mathbb{E}[\varepsilon_j^-] = 0$, and $\mathbb{E}\|\varepsilon_j^+\|_2^2 \leq \sigma^2$, $\mathbb{E}\|\varepsilon_j^-\|_2^2 \leq \sigma^2$.

Define the averaged same-modal and cross-modal noises

$$\bar{\varepsilon}_i^+ := \frac{1}{n_i^+} \sum_{j \in \mathcal{N}_i^+(f_i^*)} \varepsilon_j^+, \quad \bar{\varepsilon}_i^- := \frac{1}{n_i^-} \sum_{j \in \mathcal{N}_i^-(f_i^*)} \varepsilon_j^-.$$

Then $m_i^{same} = \mu_i + \bar{\varepsilon}_i^+$, $m_i^{cross} = \mu_i + \bar{\varepsilon}_i^-$, and hence $\tilde{z}_i - \mu_i = \alpha_i \bar{\varepsilon}_i^+ + (1 - \alpha_i)\bar{\varepsilon}_i^-$.

We first bound the second moment of $\bar{\varepsilon}_i^+$. Since the noises are mutually independent and zero mean, for $j \neq \ell$, $\mathbb{E}\langle \varepsilon_j^+, \varepsilon_\ell^+ \rangle = \langle \mathbb{E}[\varepsilon_j^+], \mathbb{E}[\varepsilon_\ell^+] \rangle = 0$. Therefore,

$$\mathbb{E}\|\bar{\varepsilon}_i^+\|_2^2 = \frac{1}{(n_i^+)^2} \mathbb{E}\left\| \sum_{j \in \mathcal{N}_i^+(f_i^*)} \varepsilon_j^+ \right\|_2^2$$

$$= \frac{1}{(n_i^+)^2} \sum_{j \in \mathcal{N}_i^+(f_i^*)} \mathbb{E}\|\varepsilon_j^+\|_2^2 + \frac{1}{(n_i^+)^2} \sum_{\substack{j,\ell \in \mathcal{N}_i^+(f_i^*) \\ j \neq \ell}} \mathbb{E}\langle \varepsilon_j^+, \varepsilon_\ell^+ \rangle$$

$$= \frac{1}{(n_i^+)^2} \sum_{j \in \mathcal{N}_i^+(f_i^*)} \mathbb{E}\|\varepsilon_j^+\|_2^2 \leq \frac{1}{(n_i^+)^2} \cdot n_i^+ \sigma^2 = \frac{\sigma^2}{n_i^+}.$$

Exactly the same argument gives $\mathbb{E}\|\bar{\varepsilon}_i^-\|_2^2 \leq \sigma^2/n_i^-$. Moreover, because $\bar{\varepsilon}_i^+$ and $\bar{\varepsilon}_i^-$ are averages of two independent zero-mean noise families, $\mathbb{E}\langle\bar{\varepsilon}_i^+, \bar{\varepsilon}_i^-\rangle = 0$. Hence,

$$
\begin{aligned}
\mathbb{E}\|\tilde{z}_i - \mu_i\|_2^2 &= \mathbb{E}\left\|\alpha_i\bar{\varepsilon}_i^+ + (1-\alpha_i)\bar{\varepsilon}_i^-\right\|_2^2 \\
&= \alpha_i^2\mathbb{E}\|\bar{\varepsilon}_i^+\|_2^2 + (1-\alpha_i)^2\mathbb{E}\|\bar{\varepsilon}_i^-\|_2^2 + 2\alpha_i(1-\alpha_i)\mathbb{E}\langle\bar{\varepsilon}_i^+, \bar{\varepsilon}_i^-\rangle \\
&= \alpha_i^2\mathbb{E}\|\bar{\varepsilon}_i^+\|_2^2 + (1-\alpha_i)^2\mathbb{E}\|\bar{\varepsilon}_i^-\|_2^2 \\
&\leq \sigma^2\left(\frac{\alpha_i^2}{n_i^+} + \frac{(1-\alpha_i)^2}{n_i^-}\right).
\end{aligned}
$$

Now let $\eta_i := \tilde{z}_i - \mu_i$. Then $\mathbb{E}[\eta_i] = 0$ and, by equation 8, $\mathbb{E}\|\eta_i\|_2^2 \leq r_i^2$.

Assume in addition that $z_i^* = \mu_i + \xi_i$, $\mathbb{E}[\xi_i] = 0$, and that $\xi_i$ is independent of all proxy noises. For a sample $k$ with $c(k) \neq c(i)$, let $\mu_k := \mu_{c(k)}$, and assume $\tilde{z}_k$ is defined analogously so that $\tilde{z}_k = \mu_k + \eta_k$, $\mathbb{E}[\eta_k] = 0$, with $\eta_k$ independent of $\xi_i$.

We first expand the within-community distance: $z_i^* - \tilde{z}_i = (\mu_i + \xi_i) - (\mu_i + \eta_i) = \xi_i - \eta_i$. Therefore,

$$
\mathbb{E}\|z_i^* - \tilde{z}_i\|_2^2 = \mathbb{E}\|\xi_i - \eta_i\|_2^2 = \mathbb{E}\|\xi_i\|_2^2 + \mathbb{E}\|\eta_i\|_2^2 - 2\mathbb{E}\langle\xi_i, \eta_i\rangle.
$$

Since $\xi_i$ is independent of the proxy noises and both $\xi_i$ and $\eta_i$ are zero mean, $\mathbb{E}\langle\xi_i, \eta_i\rangle = \langle\mathbb{E}[\xi_i], \mathbb{E}[\eta_i]\rangle = 0$. Hence $\mathbb{E}\|z_i^* - \tilde{z}_i\|_2^2 = \mathbb{E}\|\xi_i\|_2^2 + \mathbb{E}\|\eta_i\|_2^2 \leq \mathbb{E}\|\xi_i\|_2^2 + r_i^2$.

Next, for the cross-community distance,

$$
z_i^* - \tilde{z}_k = (\mu_i + \xi_i) - (\mu_k + \eta_k) = (\mu_i - \mu_k) + \xi_i - \eta_k.
$$

Thus

$$
\begin{aligned}
\mathbb{E}\|z_i^* - \tilde{z}_k\|_2^2 &= \mathbb{E}\|(\mu_i - \mu_k) + \xi_i - \eta_k\|_2^2 \\
&= \|\mu_i - \mu_k\|_2^2 + \mathbb{E}\|\xi_i\|_2^2 + \mathbb{E}\|\eta_k\|_2^2 + 2\langle\mu_i - \mu_k, \mathbb{E}[\xi_i]\rangle - 2\langle\mu_i - \mu_k, \mathbb{E}[\eta_k]\rangle - 2\mathbb{E}\langle\xi_i, \eta_k\rangle.
\end{aligned}
$$

Because $\mathbb{E}[\xi_i] = 0$, $\mathbb{E}[\eta_k] = 0$, and $\xi_i$ is independent of $\eta_k$, all three cross terms vanish. Therefore,

$$
\mathbb{E}\|z_i^* - \tilde{z}_k\|_2^2 = \|\mu_i - \mu_k\|_2^2 + \mathbb{E}\|\xi_i\|_2^2 + \mathbb{E}\|\eta_k\|_2^2 \geq \|\mu_i - \mu_k\|_2^2 + \mathbb{E}\|\xi_i\|_2^2.
$$

Subtracting the bound for $\mathbb{E}\|z_i^* - \tilde{z}_i\|_2^2$, we obtain

$$
\begin{aligned}
\mathbb{E}\|z_i^* - \tilde{z}_k\|_2^2 - \mathbb{E}\|z_i^* - \tilde{z}_i\|_2^2 &\geq \|\mu_i - \mu_k\|_2^2 + \mathbb{E}\|\xi_i\|_2^2 - \left(\mathbb{E}\|\xi_i\|_2^2 + r_i^2\right) \\
&= \|\mu_i - \mu_k\|_2^2 - r_i^2.
\end{aligned}
$$

This proves equation 9. $\qquad\square$

## C  ADDITIONAL EXPERIMENTAL RESULTS

### C.1  ADDITIONAL EXPERIMENTS FOR HASH LOOKUP

To comprehensively evaluate the performance of the proposed HINT, we present precision-recall curves, precision-N curves, and recall-N curves on the MirFlickr-25K dataset with code lengths of 32 and 64 bits. As shown in Figure 10, 9, our method consistently outperforms other approaches, which is consistent with the corresponding MAP score based on Hamming ranking. Additionally, we compute the precision and recall rates of the top-N retrieved results, demonstrating HINT's persistent advantage. Our method achieves superior performance in cross-modal hash retrieval.

### C.2  ADDITIONAL ABLATION EXPERIMENTS

To evaluate the effectiveness of these components, we introduce four variants:

- HINT V1, which only employs image-text pairs for cross-modal hash learning without the global-view consistency learning module;

Table 3: Additional ablation studies with different code lengths.

| Methods | MIRFlickr-25K | | | | | | | |
| | Image→Text | | | | Text→Image | | | |
| | 16bit | 32bit | 64bit | 128bit | 16bit | 32bit | 64bit | 128bit |
|---|---|---|---|---|---|---|---|---|
| HINT *V1* | 71.5 | 72.4 | 72.6 | 73.2 | 70.5 | 71.1 | 71.7 | 72.0 |
| HINT *V2* | 71.8 | 72.8 | 73.0 | 73.7 | 70.8 | 71.4 | 72.2 | 72.8 |
| HINT *V3* | 72.0 | 73.4 | 73.8 | 74.2 | 71.5 | 72.4 | 73.2 | 73.6 |
| HINT *V4* | 72.6 | 73.9 | 74.8 | 75.2 | 71.5 | 72.8 | 73.9 | 74.3 |
| Full Model | 72.9 | 74.4 | 75.1 | 75.5 | 72.0 | 73.1 | 74.0 | 74.9 |

Table 4: Retrieval time cost (ms) varies with code length.

| | 16 Bit | 32 Bit | 48 Bit | 64 Bit | 96 Bit | 128 Bit |
|---|---|---|---|---|---|---|
| *Hash* Code | 16.7 | 18.0 | 19.4 | 19.9 | 21.8 | 22.2 |
| *Dense* Vector | 441.4 | 491.0 | 543.0 | 602.3 | 657.7 | 696.6 |
| Speed Up | 26.5× | 27.2× | 28.0× | 30.2× | 30.1× | 31.4× |

- HINT V2, which combines image-text pairs with intra-modal KNN for cross-modal hash learning, also without the consistency learning module;

- HINT V3, which uses the average of two modalities as the learning target and builds a hierarchical encoding tree with sample selection, but without curriculum-based progressive modal alignment (Equation 6), also without the global-view consistency learning module;

- HINT V4, which adopts the full model's Lhash but excludes consistency learning.

As shown in Table 3, our results demonstrate that our complete HINT achieves optimal performance, confirming the importance of each component. Furthermore, our ablation study reveals that hierarchical encoding trees and progressive alignment yield significant improvements (*V2* and *V3*), validating our motivation.

## C.3 COMPUTATIONAL COMPLEXITY

We mainly discuss the computational complexity of the additionally introduced encoding tree construction. Assuming the number of data points is $N$ (including samples from different modalities), the time cost for calculating the similarity matrix is $O(N^2)$, the cost for constructing the KNN graph is $O(N)$, and the cost for optimizing the cross-modal encoding tree is $O(N \log^2 N)$ (Li & Pan, 2016). It is worth noting that the most time-consuming similarity matrix calculation can be accelerated by parallel computing. Moreover, the encoding tree construction is only performed once at the beginning of the training, so the additional computational complexity and time consumption brought to the overall training process are negligible.

## C.4 TIME EFFICIENCY

We conducted a speed test between HINT and dense vector retrieval on an Intel Xeon CPU E5-2697 v4 (2.30GHz), as illustrated in Table 4. The speed test is conducted on the MIRFlicker-25K dataset with a retrieval database of $10^5$ items. We perform $10^3$ runs and report the average retrieval speed cost (ms). Hash methods excel in enabling efficient and scalable image retrieval, especially for large-scale datasets, due to fast Hamming distance computation. In contrast, existing pre-trained models only output dense vectors, resulting in slower computation. Table 4 compares the efficiency of hash codes and dense codes generated by our model and a pre-trained model at various bit lengths. The results clearly demonstrate that hash codes achieve significantly faster retrieval speeds than deep feature codes, confirming their superiority, particularly in large-scale image retrieval scenarios.

Table 5: Model complexity and training efficiency comparison on MIRFlickr-25K with 128-bit.

| Method | Params (M) | Total Time (min) | MAP (%) |
|---|---|---|---|
| UCHSTM | 52.5 | 182 | 72.3 |
| UCCH | 88.0 | 58 | 73.2 |
| UDDH | 28.9 | 45 | 74.6 |
| HuggingHash+ | 40.1 | 211 | 74.5 |
| DEMO | 86.2 | 59 | 74.3 |
| HINT | 84.8 | 61 | **75.5** |

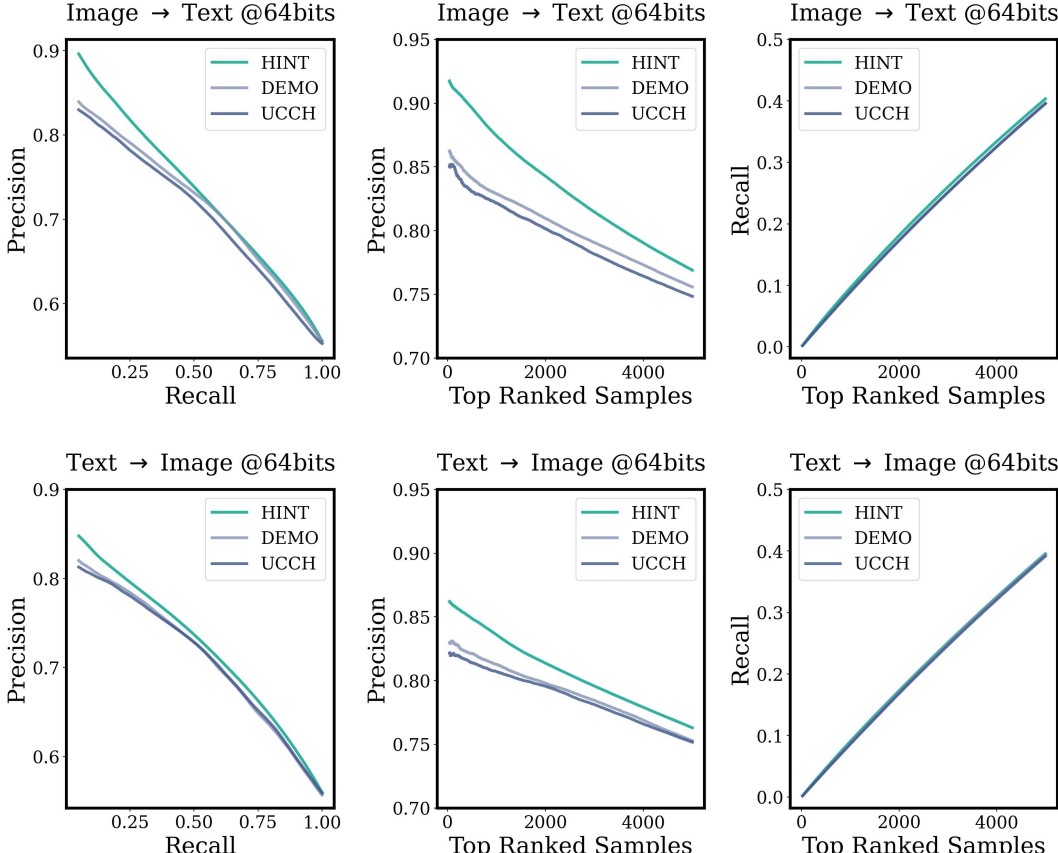

Figure 9: Hash lookup performance with 64 bits codes on the MIRFlickr-25K dataset. The precision-recall curves, precision-N curves, and recall-N curves are shown from left to right.

## C.5  COMPUTATIONAL EFFICIENCY ANALYSIS

While HINT introduces additional computational steps through the hierarchical encoding tree, we have implemented several optimizations to ensure practical efficiency:

- **Controlled Time Complexity**: The hierarchical encoding tree construction has a complexity of $O(N \log^2 N)$ and is executed only once during the initial training phase. Our experiments on MIRFlickr-25K show that tree construction takes <3 minutes (single GPU), representing <5% of total training time. Compared to existing methods like DEMO and UCCH, HINT does not significantly increase the overall training duration.

- **Memory-Efficient Design**: The encoding tree is stored using compressed relation triplets (parent-child-edge weight) instead of maintaining complete similarity matrices. Furthermore, the hash

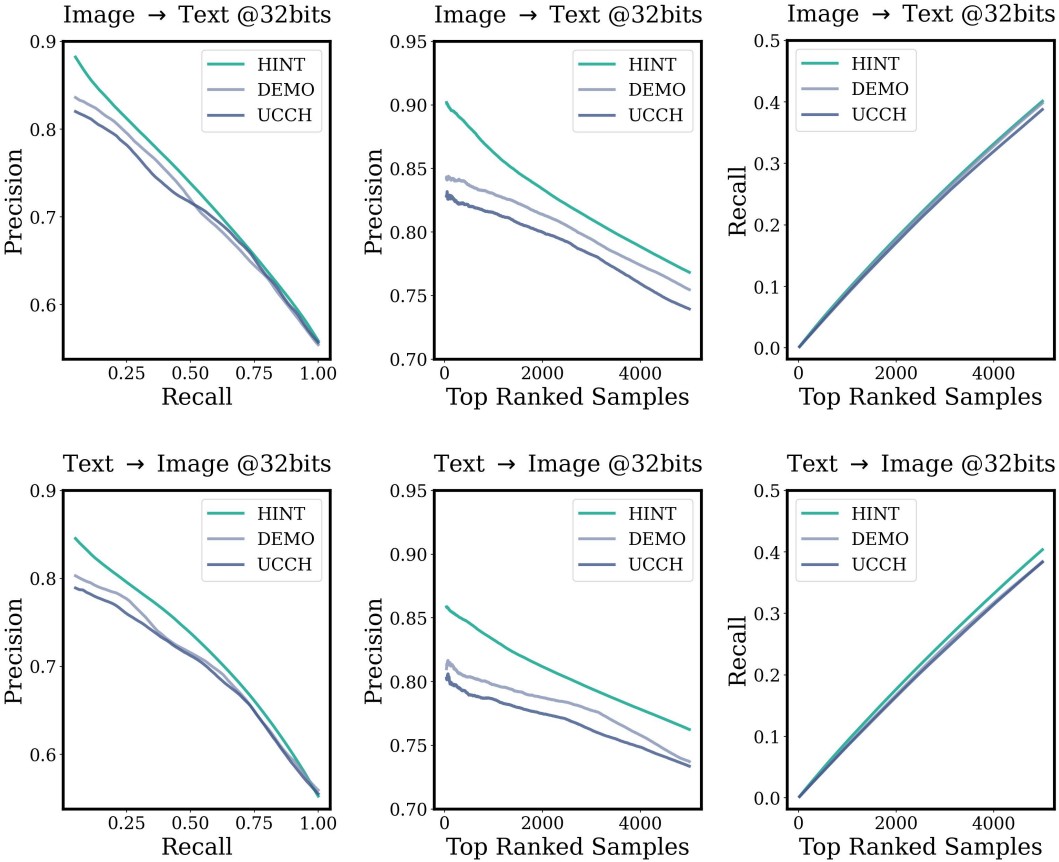

Figure 10: Hash lookup performance with 32 bits codes on the MIRFlickr-25K dataset. The precision-recall curves, precision-N curves, and recall-N curves are shown from left to right.

code generation phase is completely decoupled from the encoding tree, eliminating the need to load tree structures during inference and conserving deployment resources.

- **Practical Scalability**: The encoding tree's one-time construction and reusability make it particularly suitable for large-scale applications. This design choice significantly amortizes the initial computational investment across multiple training sessions.

These results demonstrate that HINT achieves superior performance while maintaining competitive training efficiency through its optimized design. The comprehensive model complexity and training efficiency comparison with existing methods is shown in Table 5.

### C.6 NOISE ROBUSTNESS ANALYSIS

To evaluate HINT's robustness against noisy data, we conducted experiments by randomly corrupting 10% of text-image pairs in the MIRFlickr-25K dataset. The results demonstrate HINT's superior noise resilience through both architectural design and experimental validation:

**Architectural Robustness**: HINT's hierarchical encoding tree provides two-level noise adaptation:

- The tree construction process inherently suppresses individual outliers by aggregating local semantic communities. Proxy samples, generated through neighbor feature averaging, effectively smooth out the impact of noisy samples within local communities.

- The cross-modal consistency learning module ($L_{con}$) constrains the influence of outliers on hash space mapping by enforcing semantic distribution alignment between proxy and original samples from a global perspective.

Table 6: Noise robustness comparison on MIRFlickr-25K (128 bit) with 10% corrupted pairs.

| Method | I→T | T→I | I→T($10\%n$) | T→I($10\%n$) |
|---|---|---|---|---|
| UCCH | 73.2 | 73.2 | 70.7 | 72.6 |
| DEMO | 74.3 | 74.3 | 71.4 | 73.3 |
| HINT | 75.5 | 73.8 | 74.4 | 72.9 |

Table 7: Performance comparison of different similarity metrics on MIRFlickr-25K.

| Metric | I→T | T→I |
|---|---|---|
| Cosine | 75.5 | 74.6 |
| L2 | 74.8 | 74.2 |
| L1 | 73.5 | 72.8 |

As shown in Table 6, HINT consistently maintains higher performance under noisy conditions, with minimal degradation compared to baseline methods. This demonstrates that the encoding tree's hierarchical structure effectively identifies and mitigates the interference of mismatched pairs in cross-modal alignment.

### C.7 ADDITIONAL ANALYSIS OF SIMILARITY METRICS

We conducted experiments comparing different similarity metrics for tree construction on MIRFlickr-25K. The results demonstrate that cosine similarity achieves optimal performance due to three key advantages:

- **Directional Consistency**: Cosine similarity focuses on semantic directional alignment by normalizing vector magnitudes
- **Loss Function Alignment**: The training objective relies on inner product similarity, which aligns with cosine similarity computation
- **Feature Space Compatibility**: Hamming distance is not suitable since features are not yet binarized during tree construction

The results show cosine similarity's 1.5-2.0% performance advantage over alternative metrics, confirming that feature directional alignment is more crucial than absolute distance for tree construction.

### C.8 ITERATIVE TREE CONSTRUCTION ANALYSIS

Iterative structural optimization is a promising direction (Chen et al., 2020). Our experiments reveal that a static tree construction strategy achieves better balance between efficiency and performance:

- **Empirical Results**: We compared HINT with HINT-ITER (dynamic tree updates every 5 epochs) across multiple datasets:

The results show that HINT-ITER achieves comparable but slightly lower performance while requiring more training time. Two fundamental reasons explain this phenomenon:

- **Robust Initial Structure**: Our one-time tree construction leverages hierarchical structural entropy to recover semantically coherent communities, providing a stable foundation for proxy sample generation. Iterative refinement struggles to further improve this already optimized structure.
- **Stability-Aware Alignment**: The curriculum-based mixup mechanism and consistency learning rely on stable neighborhood relationships to progressively align modalities. Frequent tree updates disrupt this process, similar to how unstable negative samples degrade contrastive learning (He et al., 2020).

While our current approach suits existing tasks, we acknowledge potential benefits of dynamic structures for specific scenarios (e.g., evolving data streams), which we leave for future work.

Table 8: Performance comparison between static and iterative tree construction.

| Method | MIRFlickr-25K | | MS-COCO | | Training |
|---|---|---|---|---|---|
| | I→T | T→I | I→T | T→I | Time (h) |
| HINT | 75.5 | 74.6 | 61.1 | 60.8 | 2.1 |
| HINT-ITER | 75.0 | 74.7 | 60.5 | 60.4 | 2.9 |

Table 9: Stability analysis of HINT across different code lengths on MIRFlickr-25K dataset. Results show mean MAP scores $\pm$ standard deviation over five runs.

| Task | 32 bits | 64 bits | 96 bits | 128 bits |
|---|---|---|---|---|
| I→T | 72.9±0.81 | 74.4±0.99 | 75.1±0.92 | 75.5±0.88 |
| T→I | 72.0±0.75 | 73.1±0.75 | 74.0±0.72 | 74.6±0.96 |

## C.9 STABILITY ANALYSIS

To rigorously assess the stability of HINT, we conducted extensive experiments with five independent runs using different random seeds. Table 9 presents the mean performance and standard deviations across different code lengths for both Image-to-Text (I→T) and Text-to-Image (T→I) retrieval tasks on the dataset.

The results demonstrate that HINT maintains consistent performance with remarkably low variance across different code lengths. The standard deviations consistently remain below 1% for both retrieval directions, indicating strong robustness to random initialization. This stability can be attributed to our hierarchical encoding tree structure and curriculum-based progressive alignment strategy, which provide reliable guidance for hash code learning regardless of initialization conditions.

## C.10 TAIL CLASS PERFORMANCE ANALYSIS

To evaluate HINT's robustness under class imbalance, we conducted tail class analysis on MIRFlickr-25K dataset. We categorized the 24 classes into three groups based on sample counts: Head (8 classes), Medium (10 classes), and Tail (6 classes). As shown in Figure 11, HINT demonstrates consistent advantages across all class groups, with particularly significant improvements on Tail classes. This superior performance can be attributed to HINT's hierarchical structure: the encoding tree leverages cross-modal connections to augment sparse intra-modal neighborhoods for tail classes, while the hierarchical community structure ensures these connections remain semantically coherent rather than introducing noise from head classes.

## D ADDITIONAL DISCUSSION

### D.1 RATIONALE FOR HIERARCHICAL MINING OF RELATIONSHIPS

Hierarchical semantic structures are inherent in real-world data. Visual and textual content naturally form multi-level conceptual taxonomies. For instance, a general category like "Objects" can be decomposed into "Animals" and "Vehicles". "Animals" can be further subdivided into "Domestic" and "Wild", with "Domestic" containing specific instances like "Cats" and "Dogs" (e.g., "Maine Coon", "Siamese"). Similarly, "Vehicles" might branch into "Ground" and "Air" transport, with "Ground" including "Cars" and "Trucks".

Prevailing unsupervised cross-modal hashing methods often rely on flat representations of image-text pair relationships. Such flat structures exhibit limitations in capturing these intrinsic hierarchical dependencies. Specifically, they may:

- Treat the semantic dissimilarity between disparate pairs (e.g., "Cat-Dog" vs. "Cat-Car") undifferentiatedly, failing to recognize varying degrees of relatedness based on hierarchical proximity.

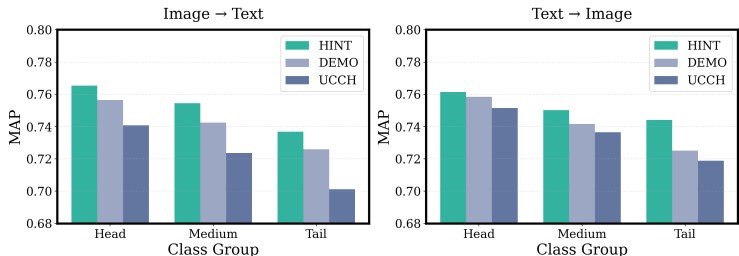

Figure 11: Performance comparison on Head, Medium, and Tail class groups on MIRFlickr-25K (128-bit). HINT demonstrates consistent advantages across all groups, with more significant improvements on Tail classes compared to baselines.

- Implicitly assume transitive relationships, an assumption that does not consistently hold for complex semantic relationships across different levels of abstraction.
- Struggle to ensure that instances within a sub-category (e.g., "Maine Coon" and "Siamese" under "Cats") are represented as being semantically closer to each other than to instances from distant categories (e.g., "Cars").

Our proposed HINT addresses these limitations through the Hierarchical Encoding Tree, which explicitly discovers and models inherent semantic hierarchies by optimizing structural entropy. This hierarchical approach offers several advantages:

- It facilitates a deeper exploration of semantic relationships that extend beyond direct, observed pairings, uncovering latent community structures.
- It enables the generation of more generalizable hash codes that are grounded in these discovered semantic communities, rather than isolated instances.
- It promotes smoother and more effective cross-modal alignment through the use of proxy samples derived from semantic neighborhoods and a curriculum learning strategy, thereby more effectively bridging the semantic gap between modalities.

By capturing these multi-level containment relationships, HINT can learn hash codes that better reflect real-world semantic structures, leading to improved retrieval performance. Current flat modeling approaches, by contrast, which treat "Cat-Dog" and "Cat-Car" similarity differences with equal weight, miss this crucial hierarchical semantic information, significantly impeding their ability to learn generalizable hash codes that align with complex real-world semantic organizations.

# E  ADDITIONAL IMPLEMENTATION DETAILS

## E.1  BASELINE DETAILS

The introduction of the baseline methods is as follows:

- **CVH** (Kumar & Udupa, 2011) transforms the learning process into a manageable feature-based hashing problem.
- **LSSH** (Zhou et al., 2014) is an effective iterative strategy, which explores the correlations between multi-modal representations and bridges the semantic gaps in the latent semantic space.
- **CMFH** (Ding et al., 2016a) exploites cross-modal decomposition to establish strong connections.
- **FSH** (Liu et al., 2017) alternates optimization to learn consistent binary codes and hash functions.
- **MTFH** (Liu et al., 2019a) produces modality-specific binary codes with varying lengths while ensuring the comparability of heterogeneous data.
- **FOMH** (Lu et al., 2019) integrates a self-weighted and flexible multi-modal fusion strategy, enabling robust fusion even when missing modalities.
- **DCH** (Xu et al., 2017) utilizes an efficient optimization algorithm to produce the optimized bit recursively.

- **DGCPN** (Yu et al., 2021) is a graph neighborhood approach, which explore the relationships between data points and their neighbors through a graph neighborhood approach, thereby enhancing the accuracy of data similarity measurement.

- **UCHSTM** (Tu et al., 2023) employs a custom-designed similarity loss to rectify any ill-defined similarities in the instance similarity matrix.

- **UCCH** (Hu et al., 2022) uses contrastive learning to enforce alignment between different modalities and unified binary representations, focusing on leveraging discriminative information from all pairs.

- **UDDH** (Zhang et al., 2024a)is a dual deep hashing architecture that combines semantic indexing with content codes for cross-modal retrieval.

- **HuggingHash+** (Wang et al., 2024b) is a transformer-based multi-granularity learning framework for unsupervised cross-modal hashing.

- **DEMO** (Zhang et al., 2024b) utilizes multi-view augmentation to represent each image, followed by parameterized distribution divergence to ensure robust similarity structures.

- **GCRH** (Wei et al., 2025) is a contrastive-and-reconstructive framework combining global graph contrastive learning with local graph reconstruction to reduce the modality gap.

- **VTM-UCH** (Fan & Cao, 2025) utilizes vision-guided text mining by combining vision-language models with community detection for refining hash code distribution.

### E.2   DATASET DETAILS

We conduct experiments on three widely used public datasets:

- *MIRFlickr-25K* (Huiskes & Lew, 2008) contains $25,000$ text-image pairs. Each text sample is embedded using the Bag-of-Words (BoW) strategy.

- *NUS-WIDE* (Rasiwasia et al., 2010) comprises $269,498$ text-image pairs with multiple labels from $81$ categories, where each text is embedded into a 1000-dimensional vector via BoW.

- *MS-COCO* (Lin et al., 2014) includes $123,287$ text-image pairs with multiple labels from $80$ categories. Each text sample is embeded into a 2026-dimensional BoW vector.

Following the problem settings of the latest baseline (Zhang et al., 2024b), each dataset is divided into a query set and a retrieval set. During the training process, only text-image pair information is accessible without label information.

### E.3   IMPLEMENTATION DETAILS

Our method is implemented based on the latest baseline (Zhang et al., 2024b). For training, we maintain consistency with baseline approaches by selecting 10,000 samples as our training set. We utilize pre-extracted visual and text feature vectors, which are mapped to the Hamming space through two-layer MLPs with a dimension of 512. The implementation is done using PyTorch framework, with all experiments conducted on a single NVIDIA A40 GPU.

### E.4   EVALUATION METRICS

Mean Average Precision (MAP) is a comprehensive metric widely used to evaluate retrieval performance in cross-modal hashing research (Wang et al., 2010; Liu et al., 2012; Shen et al., 2015). The MAP score has a range of $0$ to $1$, where higher values indicate better retrieval performance. It works by calculating the average precision for each query, followed by taking the mean across the queries in the test set. This provides a single-figure measure that reflects system performance across all relevant documents and all recall levels. MAP considers both precision and recall aspects of the retrieval system, making it particularly suitable for evaluating hashing-based retrieval systems where we are concerned with the overall ranking quality of results.

## F  THE USE OF LARGE LANGUAGE MODELS

In this article, we use LLM for language polishing and retrieval of the latest research works. We confirm that we take full responsibility for the contents written in this paper.

