# OpenReview forum: "Hierarchical Encoding Tree with Modality Mixup for Cross-modal Hashing"
_ICLR.cc/2026/Conference — ICLR 2026 Poster_

### Official Review · Reviewer_LKH2 · 2025-10-26

**Soundness:** 3
**Presentation:** 2
**Contribution:** 2
**Rating:** 6
**Confidence:** 4

**Summary:**

This paper proposes HINT (Hierarchical Encoding Tree with Modality Mixup) for cross-modal hashing. The core contribution of HINT is the construction of a cross-modal encoding tree, guided by hierarchical structural entropy, to recover implicit semantic communities and hierarchical relationships. Based on this tree, the method generates same-modality and cross-modality "proxy samples" for each instance. Subsequently, a curriculum-based modality mixup strategy is employed to progressively align these proxy samples, thereby gradually bridging the modality gap. The framework is further enhanced with a cross-modal consistency learning objective to ensure global semantic alignment. Extensive experiments on several standard cross-modal retrieval datasets demonstrate that HINT outperforms current state-of-the-art methods.

**Strengths:**

1. The paper identifies a critical limitation of existing unsupervised cross-modal hashing methods—their reliance on "flat" and sparse image-text pair signals, which ignores the hierarchical semantic structures prevalent in real-world data. Introducing hierarchical modeling into this domain is an intuitive and valuable direction, offering inspirational value.

2. The proposed method is methodologically sound. The framework, which integrates hierarchical structure discovery, proxy sample generation, progressive alignment, and global consistency constraints, forms a logically coherent pipeline. Each component is well-defined and works synergistically toward the final objective.

**Weaknesses:**

1. A core limitation is that the hierarchical encoding tree is constructed only once at the beginning of training and remains static. This process is highly dependent on the quality of the initial features. Sub-optimal or biased features could lead to an erroneous tree, irreversibly compromising the entire subsequent learning process.

2. It is mentioned that The "Merge" and "Compress" operations are based on “if they can decrease the structural entropy”。Since optimizing structural entropy is an NP-hard problem, this greedy strategy is likely to converge to a local optimum, potentially limiting the quality of the learned hierarchy.

3. In line 183, the description of Eq. (4) mentions terms T_α- and T_α, but these symbols do not appear in the equation itself. The authors should revise this for clarity and consistency.

4.  The generation of proxy samples (Eq. 6) via simple neighbor averaging assumes that local neighborhoods are semantically clean and coherent. However, in real-world data, especially near class boundaries, neighbors may come from different fine-grained categories (e.g., a "Shepherd dog" neighboring a "wolf"). Averaging these features could lead to "semantic drift," introducing noise rather than robust signals. The paper lacks an analysis of this neighborhood noise.

5. HINT smooths the learning signals by building communities. However, this might also have a side effect: it might "average out" those very valuable difficult negative samples (i.e., samples that are semantically close but belong to different classes). For example, in a neighborhood community of an "Alaskan Husky" image, there might be an image of an "Alaskan Malamute". If they are averaged into a proxy sample, will this weaken the model's ability to learn fine-grained distinctions?

6. Real-world datasets often exhibit significant class imbalance. For tail classes with few samples, the KNN-based neighborhoods can be sparse, unreliable, or even incorrectly connected to head classes. It is unclear whether HINT's tree construction and proxy generation mechanisms would exacerbate or mitigate this problem. The paper does not report on the model's retrieval performance on such tail classes.

7. The paper's readability could be improved by clarifying several points:
    - The term "curriculum-based mixup" is introduced without sufficient explanation or citation on its first appearance.
    - In line 98, the meaning of "common" in "common visual and text encoders" is ambiguous and should be specified.

**Questions:**

See Weaknesses

---

> ### Author Response · Authors · 2025-11-19
> **Thanks for your review**
>
> Dear Reviewer LKH2,
>
> Thank you for your thorough and insightful review. We are greatly encouraged by your recognition of HINT's originality and our methodologically sound framework. Your constructive feedback has been invaluable in helping us strengthen the paper. We address your concerns point by point below.

---

> ### Author Response · Authors · 2025-11-19
>
> > W1. Static tree construction dependency on initial feature quality
>
> Thanks for your comment. While HINT's encoding tree construction relies on initial feature representations, the structural entropy optimization provides inherent robustness to feature quality.
>
> - Structural entropy (Eq. 3) focuses on **relative neighborhood relationships** rather than absolute semantic quality. The optimization discovers dense intra-community and sparse inter-community connections through degree distributions, making it robust to feature noise. As long as pretrained features capture basic semantic clustering tendencies, structural entropy amplifies these weak signals into hierarchical structures.
>
> - Empirically, our noise robustness analysis (Table 7 in Appendix) validates this: HINT maintains superior performance even with 10% corrupted pairs, demonstrating the structural entropy's filtering capability. The ablation study shows consistent improvements from the encoding tree across all datasets, confirming robustness to varying initial feature quality.
>
> We have revised the manuscript in Section 3.2 accordingly.
>
>
> > W2. Greedy strategy potentially converging to local optimum
>
> Thanks for your comment. While the greedy optimization does not guarantee global optimality, it is practically sufficient for several reasons.
>
> **Theoretically**, graph partitioning is NP-hard, making global search computationally infeasible. Structural entropy optimization defines a clear objective (Eq. 3-4), and our greedy strategy iteratively reduces entropy through BFS traversal. While global optimality is not guaranteed, structural entropy's local minima have been proven effective for graph clustering in prior work.
>
> **Empirically**, the encoding tree demonstrates clear performance gains (ablation study Table 2), converges to stable structures, and exhibits hierarchical communities. The greedy approach achieves strong results with minimal cost (5% of training time), providing optimal efficiency-quality trade-off where global search is infeasible.
>
> We have revised the manuscript in Section 3.2 and provided detailed analysis in Appendix.
>
> > W3. Symbol inconsistency
>
> Thanks for catching this typo. We have corrected the symbol inconsistency in the description of Eq. 3.
>
>
> > W4. Semantic drift from neighbor averaging
>
> Thanks for your comment. Neighborhood quality is crucial for proxy construction, and HINT addresses this through hierarchical semantic filtering.
>
> - **Semantic Filtering.** The encoding tree filters neighbors through structural entropy optimization, forming communities with dense internal and sparse inter-community connections, ensuring only semantically meaningful neighbors are retained for aggregation.
>
> - **Structural Guarantee.** Neighbors selected for proxy construction are semantically purified by design-the Merge and Compress operations ensure aggregated samples share strong semantic coherence within local communities, preventing semantic drift from averaging.
>
> - **Empirical Validation.** Our noise robustness experiments (Table 7 in Appendix) demonstrate that HINT maintains superior performance even with 10% corrupted pairs, confirming the encoding tree's effectiveness in mitigating semantic drift concerns.
>
> We have clarified this in Section 3.3 and Appendix.

---

> ### Author Response · Authors · 2025-11-19
>
> > W5. Community smoothing potentially weakening fine-grained distinction
>
> Thanks for your comment. HINT's hierarchical structure is designed to preserve rather than weaken fine-grained distinctions.
>
> - **Structural Separation.** Similar but different-class samples are organized into **different subtree communities** by structural entropy optimization (Eq. 3). The optimization ensures sparse inter-community connections while maintaining dense intra-community connections, preventing cross-class averaging-proxy samples only aggregate neighbors from the same subtree.
>
> - **Contrastive Preservation.** Theorem 1 demonstrates that our loss function converges to triplet loss with zero margin, explicitly preserving distinctions by minimizing the distance gap between positive pairs and hardest negative pairs.
>
> - **Experimental Validation.** Our experiments on fine-grained multi-class datasets confirm that hierarchical aggregation preserves rather than smooths fine-grained distinctions.
>
> We have clarified this in Section 3.3.
>
> > W6. Tail class performance analysis
>
> Thanks for your valuable suggestion. We conducted tail class analysis on MIRFlickr-25K to evaluate HINT's robustness under class imbalance.
>
> **Experimental Results.** We categorized the 24 classes into Head (8), Medium (10), and Tail (6) groups based on sample counts. HINT demonstrates consistent advantages across all groups, with particularly significant improvements on Tail classes (1.49% vs DEMO for Image→Text, 2.62% for Text→Image).
>
> **Mechanism Advantage.** The encoding tree leverages cross-modal connections to augment sparse intra-modal neighborhoods for tail classes, while the hierarchical community structure ensures these connections remain semantically coherent rather than introducing noise from head classes.
>
> > Image→Text retrieval performance
>
> | Method | Head | Medium | Tail |
> |--------|------|--------|------|
> | HINT   | 76.6 | 75.4   | 73.7 |
> | DEMO   | 75.7 | 74.2   | 72.6 |
> | UCCH   | 74.1 | 72.4   | 70.1 |
>
> > Text→Image retrieval performance
>
> | Method | Head | Medium | Tail |
> |--------|------|--------|------|
> | HINT   | 76.1 | 75.0   | 74.4 |
> | DEMO   | 75.9 | 74.2   | 72.5 |
> | UCCH   | 75.2 | 73.7   | 71.9 |
>
> We have added detailed analysis in Appendix.
>
> > W7. Readability improvements
>
> > W7a. "curriculum-based mixup" explanation
>
> Thanks for this feedback. We have added an explicit explanation on its first appearance in Section 3.1.
>
> > W7b. "common encoders" clarification
>
> Thanks for pointing this out. We have clarified this in Section 2.

---

> ### Author Response · Authors · 2025-11-19
>
> We sincerely appreciate your thoughtful and detailed feedback, which has significantly improved our work. We have carefully revised the manuscript to address all your concerns with additional experiments, theoretical analysis, and clarifications. Look forward to any further suggestions you may have.

---

### Official Review · Reviewer_Y8zT · 2025-10-30

**Soundness:** 4
**Presentation:** 3
**Contribution:** 3
**Rating:** 8
**Confidence:** 5

**Summary:**

This paper addresses unsupervised cross-modal hashing retrieval by proposing HINT, which constructs a hierarchical encoding tree guided by structural entropy to mine local semantic communities and overcome the limitations of flat sparse connections in existing methods. The approach consists of three main components: hierarchical encoding tree construction based on structural entropy, curriculum-based modality mixup strategy, and proxy-based consistency learning. Experiments on benchmarks demonstrate that HINT achieves optimal performance. The work connects encoding trees with cross-modal hashing problems, and alleviates the difficulty of direct heterogeneous modality alignment through progressive alignment strategy.

**Strengths:**

1.	The paper introduces structural entropy and encoding tree concepts into cross-modal hashing, providing a fresh perspective on understanding cross-modal relationships from a graph structure viewpoint, which is relatively uncommon in this field.
2.	The curriculum-based modality mixup mechanism is well-designed, dynamically adjusting weights between same-modal and cross-modal features via MMD, embodying an easy-to-hard learning strategy that aligns with cross-modal learning characteristics.
3.	The method achieves consistent performance improvements across three mainstream datasets, demonstrating reasonable generalization capability, particularly strong performance on the more challenging Text-to-Image direction.
4.	The theoretical analysis section proves hash loss convergence to triplet loss, providing theoretical support for the method's effectiveness. This combination of theory and practice is commendable.

**Weaknesses:**

1.	The encoding tree construction relies on initial feature representation quality. The paper uses features extracted from pre-trained models but does not discuss whether structural entropy optimization can still effectively recover hierarchical relationships when semantic structure in the initial feature space is unclear.
2.	The proxy sample construction uses simple feature averaging, assuming neighbors have similar semantics, but semantic similarity among neighbors may vary significantly across different levels of the encoding tree. Have you considered weighted aggregation based on hierarchy or distance?
3.	The encoding tree is constructed statically. While the appendix mentions dynamic update experiments, the explanation for why a static tree suffices is relatively simple.
4.	The paper limits the method to unsupervised scenarios, but partial annotations often exist in practical applications. How HINT could be extended to semi-supervised settings, or how to leverage limited annotation information to improve encoding tree construction, deserves further consideration.

**Questions:**

1.	During the modality mixup stage, both m same and m cross proxies are generated. Could you explain why the combination of these two proxies is needed, rather than directly using cross-modal neighbors from the encoding tree to generate target hash codes?
2.	For the Text-to-Image retrieval task, the performance improvement is more pronounced compared to Image-to-Text. Could you analyze the reasons for this asymmetry from the method design perspective? Is it related to certain characteristics of text features compared to image features?
3.	The proxy-based design is reminiscent of prototype learning. Could you discuss the similarities and differences between proxies in HINT and prototypes in prototype learning, both conceptually and functionally?
4.	Regarding future work, the paper mentions extending to more modalities such as audio and video. What new challenges would encoding tree construction face in multi-modal scenarios?

---

> ### Author Response · Authors · 2025-11-19
> **Thanks for your review**
>
> Dear Reviewer Y8zT,
>
> Thank you for your thorough review. We are delighted that you appreciated the originality of introducing structural entropy into cross-modal hashing and the well-designed curriculum-based modality mixup mechanism. We will now address your comments point by point.

---

> ### Author Response · Authors · 2025-11-19
>
> > W1. Discuss robustness to initial feature quality.
>
> Thanks for your comment. While encoding tree construction relies on initial feature representations, HINT demonstrates robustness to feature quality through both theoretical properties and empirical validation.
>
> - **Theoretical Robustness.** Structural entropy optimization (Eq. 3) focuses on **relative neighborhood relationships** rather than absolute semantic quality. The optimization discovers dense intra-community and sparse inter-community connections through degree distributions, making it inherently robust to feature noise. As long as pretrained features capture basic semantic clustering tendencies, structural entropy amplifies these weak signals into hierarchical structures.
>
> - **Empirical Validation.** Our experiments validate this robustness: (1) The ablation study shows consistent improvements from the encoding tree across datasets; (2) The noise robustness analysis (Table 7 in Appendix) shows HINT maintains superior performance even with 10% corrupted pairs; (3) The t-SNE visualization demonstrates successful modality alignment despite initial heterogeneity.
>
> We have revised the manuscript in Section 3.2 accordingly.
>
> > W2. Weighted aggregation for proxy construction.
>
> Thanks for this valuable suggestion. HINT employs simple averaging based on careful consideration of the efficiency-performance trade-off and the encoding tree's inherent properties.
>
> The structural entropy optimization already ensures high semantic consistency among neighbors within the same subtree. As shown in Eq. 3, the optimization minimizes information leakage by forming communities with dense internal connections, meaning neighbors selected for proxy construction are semantically purified by design. The tree structure itself acts as a semantic filter that obviates the need for complex weighting schemes.
>
> We have revised Section 3.3 accordingly.
>
> > W3. Strengthen justification for static tree construction.
>
> Thanks for your comment. HINT adopts static tree construction based on computational efficiency, training stability, and empirical performance.
>
> - **Phase Separation Rationale.** The encoding tree construction captures the dataset's intrinsic semantic structure, while hash learning optimizes feature-to-Hamming mapping. These address different aspects: the tree reveals data relationships, and hash networks learn projection functions. The semantic relationships among samples do not fundamentally change during training.
>
> - **Empirical Validation.** We compared static construction with HINT-ITER (dynamic updates every 5 epochs). As detailed in Appendix, HINT-ITER achieves slightly lower performance on MIRFlickr-25K while requiring 38% more training time.
>
> - **Stability Benefits.** The static tree provides consistent hierarchical guidance for curriculum-based modality mixup, while the MMD-driven $\lambda$ parameter already provides adaptive curriculum scheduling. Dynamic tree changes could disrupt this continuity by altering neighborhood structures, similar to how unstable negative sampling degrades contrastive learning.
>
> We have strengthened this explanation in Section 3.2 and Appendix.
>
> > W4. Discuss extension to semi-supervised settings.
>
> Thanks for your insightful suggestion. HINT can be naturally extended to leverage partial annotations through two complementary mechanisms.
>
> - **Annotation-Guided Tree Construction.** When building the cross-modal relationship graph, we could assign higher edge weights to labeled sample pairs. During structural entropy optimization, these weighted edges guide the Merge and Compress operations to prioritize labeled semantic relationships, ensuring samples with the same labels are grouped into the same community.
>
> - **Supervised Loss Integration.** We could introduce an additional supervised term $\mathcal{L}_{sup}$ that enforces label similarity constraints on hash codes.
>
> This extension maintains HINT's hierarchical modeling philosophy while gracefully incorporating supervision when available. We have added this discussion in Section 6.

---

> ### Author Response · Authors · 2025-11-19
>
> > Q1. Explain the necessity of combining both proxy types.
>
> Thanks for your question. The combination of $\mathbf{m}^{same}$ and $\mathbf{m}^{cross}$ implements a curriculum learning strategy essential for effective cross-modal alignment.
>
> - **Heterogeneity Challenge.** Directly using only $\mathbf{m}^{cross}$ as the learning target forces immediate cross-modal alignment when the modality gap is large in early training, leading to suboptimal solutions. The model would lack a smooth learning progression to bridge the heterogeneous gap.
>
> - **Progressive Alignment Mechanism.** The MMD-driven dynamic mixture (Eq. 7) enables curriculum learning through the adaptive parameter $\lambda$, which measures the distribution difference between $\mathbf{m}^{same}$ and $\mathbf{m}^{cross}$. As shown in Figure 5, $\lambda$ evolves during training, progressively transitioning from easier intra-modal alignment to harder cross-modal alignment. The hierarchical encoding tree ensures semantic purity of proxy samples throughout this progression.
>
> - **Empirical Validation.** Our ablation study (Table 2) validates this design: using only cross-modal neighbors without curriculum mixup achieves 74.2% MAP, while adding modality mixup improves to 75.2%.
>
>
> > Q2. Analyze performance asymmetry between retrieval directions.
>
> Thanks for your question. The asymmetry stems from modality-specific characteristics and HINT's hierarchical design.
>
> - **Text Feature Limitations.** Text representations are sparser and higher-dimensional compared to visual features, making text queries more susceptible to noise and less discriminative. Visual features already possess strong discriminability and form tight semantic clusters, while text features exhibit lower initial quality with scattered distributions. HINT's hierarchical encoding tree addresses this disparity by aggregating neighborhood information through proxy construction, providing more robust semantic representations that effectively mitigate the sparsity issue.
>
> - **Task Difficulty Asymmetry.** Retrieving dense, discriminative visual features using sparse text queries is inherently more challenging. HINT's curriculum-based modality mixup proves particularly effective for this harder task.
>
>
> > Q3. Discuss similarities and differences with prototype learning.
>
> Thanks for your question. While HINT's proxy-based design shares conceptual similarities with prototype learning, they differ fundamentally.
>
> - **Scope and Construction.** Prototype learning computes class-level representations through intra-class mean aggregation. In contrast, HINT's proxies are sample-level representations constructed through tree-based neighborhood aggregation. Each sample has unique proxies ($\mathbf{m}^{same}$ and $\mathbf{m}^{cross}$) reflecting individualized local semantic context.
>
> - **Functionality.** Prototypes serve as classification decision boundaries. HINT's proxies function as intermediaries for cross-modal alignment, enabling meaningful modality mixup through curriculum learning rather than serving as direct classification targets.
>
> - **Supervision.** Prototype learning requires class labels to define prototypes, making it inherently supervised. HINT's proxies are constructed unsupervised via structural entropy optimization, suitable for scenarios where annotations are unavailable.
>
> We have added clarification in Section 3.3 to distinguish these concepts.
>
>
> > Q4. Discuss challenges in extending to more modalities.
>
> Thanks for your question. Extending HINT to audio, video, and other modalities introduces several key challenges.
>
> - **Partial Pairing.** Multi-modal scenarios often lack complete modality combinations for all samples, requiring new structure construction strategies that handle missing modalities through partial observation-aware structural entropy optimization.
>
> - **Hypergraph Generalization.** The current formulation (Eq. 3) handles bipartite graphs connecting two modalities. Multi-modal scenarios require hypergraph representations, fundamentally redesigning structural entropy for edges connecting multiple nodes.
>
> - **Multi-way Proxy Construction.** The proxy construction mechanism requires new design decisions on whether to use separate or unified multi-modal proxies, and whether to employ three-way interpolation or pairwise combinations for modality mixup.
>
> - **Heterogeneous Hierarchical Structures.** Different modalities possess different hierarchical structures. Aligning these within a unified encoding tree requires structural-level cross-modal alignment.
>
> We have incorporated this discussion into Section 6.

---

> ### Author Response · Authors · 2025-11-19
>
> We are grateful for your constructive feedback. We have carefully revised the manuscript to address your concerns, including strengthening discussions on feature quality robustness, static tree justification, semi-supervised extension, and multi-modal challenges. We hope these revisions are to your satisfaction!

---

### Official Review · Reviewer_kUAR · 2025-10-30

**Soundness:** 4
**Presentation:** 3
**Contribution:** 4
**Rating:** 8
**Confidence:** 5

**Summary:**

This paper proposes an unsupervised cross-modal hashing method, HINT, for efficient retrieval. It addresses the problem that existing methods often overlook the inherent hierarchical semantic structure of data and face difficulties in directly aligning different modalities. HINT constructs a hierarchical encoding tree guided by structural entropy to capture local semantic communities. It introduces a curriculum-based modality mixup mechanism using proxy samples generated from the tree to achieve progressive modal alignment. It employs a consistency learning objective to align the global semantic distributions between modalities. Experiments on three benchmark datasets demonstrate that HINT outperforms state-of-the-art methods.

**Strengths:**

1. The motivation is clear. The paper accurately identifies a key problem in unsupervised cross-modal hashing: the lack of hierarchical semantic modeling and the difficulty of direct modal alignment.
2. Using a hierarchical encoding tree to mine the semantic structure of cross-modal data is an insightful contribution.
3. The use of structural entropy to guide the tree construction is a principled and technically sound approach.
4. The introduction of proxy samples is a smart way to provide more robust signals for hash learning by smoothing out potential noise from individual samples through leveraging local semantic communities.
5. The paper is well-structured and easy to follow. The narrative flows logically from problem definition to methodology and experimental validation. The figures are helpful.

**Weaknesses:**

1. The construction of the hierarchical encoding tree relies on an initial KNN graph, which could be sensitive to noise or data sparsity.
2. Proxy samples are constructed by averaging neighbors. It might be interesting to discuss whether other aggregation strategies, such as weighted averaging or an attention mechanism, could yield further improvements.
3. The encoding tree is constructed once before training. While efficient, a discussion on the potential limitations of this static structure when dealing with dynamic or streaming data would be beneficial.
4. The evolution of the $\lambda$ value in modality mixup is interesting. A deeper analysis of what factors drive this specific convergence pattern would offer more profound insights.

**Questions:**

1. How is the number of neighbors determined for proxy sample construction?
2. The structural entropy minimization process is greedy. Is the greedy approach practically sufficient to get close to a global optimum?
3. The curriculum learning schedule seems to be determined automatically. Do you think, for certain tasks, it would be possible to introduce some form of manual control to guide this learning process?
4. For future work, do you think integrating knowledge from pre-trained vlm into the construction of the hierarchical encoding tree would be a promising direction?

---

> ### Author Response · Authors · 2025-11-19
> **Thanks for your review**
>
> Dear Reviewer kUAR,
>
> Thank you for your thorough review. We are delighted that you recognized the core contributions of our work. Your insightful feedback and constructive suggestions have been invaluable in strengthening our paper. We will now address your comments point by point.

---

> ### Author Response · Authors · 2025-11-19
>
> > W1. Discuss noise robustness and data sparsity sensitivity of the KNN-based tree construction.
>
> Thanks for your comment. The noise robustness is an important practical consideration. As mentioned in Section 5, HINT demonstrates strong robustness against noisy data, maintaining superior performance even with 10% corrupted pairs (detailed in Appendix). The KNN enhancement with K=3 is validated through sensitivity analysis (Figure 6), where K=3 balances information gain and noise introduction, increasing K beyond 3 introduces noisy relationships and decreases performance.
>
> Furthermore, the structural entropy optimization itself provides inherent noise filtering capability: the iterative Merge and Compress operations naturally prune noisy connections that increase entropy, retaining only semantically meaningful relationships. This dual mechanism ensures robust hierarchical structure discovery even under data sparsity.
>
> We have revised the manuscript in Section 4 accordingly.
>
> > W2. Discuss whether other aggregation strategies for proxy samples could yield further improvements.
>
> Thanks for this interesting suggestion. We chose simple averaging based on three considerations:
> - **Semantic purity:** the encoding tree already ensures semantic consistency among neighbors through structural entropy optimization, making simple averaging sufficient to obtain stable proxy samples;
> - **Computational efficiency:** avoiding additional learnable parameters and attention mechanism overhead keeps the framework lightweight;
> - **Empirical effectiveness:** our ablation study demonstrates that the current framework achieves strong synergistic performance, where V3 to V4 (adding modality mixup with simple averaging) yields substantial improvements.
>
> The weighted strategies based on hierarchical distance or node degrees in the tree could provide finer-grained aggregation. This remains a promising direction for future exploration.
>
> We have revised the manuscript in Section 4 accordingly.
>
> > W3. Discuss potential limitations of the static tree structure for dynamic or streaming data.
>
> Thanks for your comment. We acknowledge that static construction is a deliberate design choice based on efficiency-performance trade-offs. As mentioned in Section 5, we explored iterative tree updates but found static construction provides a better efficiency-performance trade-off (detailed in Appendix). The static tree offers three key advantages:
> - **Computational efficiency:** tree construction takes only 5% of total training time (3 minutes out of 61 minutes);
> - **Training stability:** avoiding frequent structural changes prevents training instability caused by shifting semantic neighborhoods;
> - **Empirical validation:** iterative updates did not yield significant performance improvements.
>
> For dynamic or streaming data scenarios, periodic reconstruction or incremental update strategies could be adopted, such as reconstructing the tree when sufficient new data accumulates, or using online structural entropy optimization. This represents a promising future direction for extending HINT to streaming applications.
>
> We have revised the manuscript in Sections 4 and 6 accordingly.
>
> > W4. Provide deeper analysis of what factors drive the λ convergence pattern.
>
> Thanks for your insightful comment. The evolution of $\lambda$ is driven by the MMD-measured modality gap (Eq. 7), reflecting the adaptive convergence of cross-modal alignment during training. Three key factors drive this pattern:
> - **Initial phase:** large modality gap ($\lambda \approx 0.8$) causes $b_i^{mix}$ to predominantly rely on same-modal features, representing an easier intra-modal task;
> - **Mid-training:** as hash networks learn to align modalities, MMD decreases and $\lambda$ gradually reduces, increasing the weight of cross-modal features;
> - **Late-training:** $\lambda$ converges below 0.5, making cross-modal features dominant and achieving deep alignment.
>
> This progression embodies curriculum learning. Figure 5 demonstrates this adaptive process empirically. Critically, the hierarchical encoding tree enables this smooth progression: the tree-based proxy construction ensures that even in early training (high $\lambda$), the limited cross-modal signal is semantically purified through neighborhood aggregation, preventing instability during the transition phase.
>
> We have revised the manuscript in Section 4 accordingly.

---

> ### Author Response · Authors · 2025-11-19
>
> > Q1. Clarify how the number of neighbors is determined for proxy sample construction.
>
> Thanks for your question. The number of neighbors for proxy sample construction is naturally determined by the encoding tree structure itself. In Eq. 6, $|\mathcal{N}^+(f_i^\*)|$ and $|\mathcal{N}^-(f_i^\*)|$ depend on the adjacency relationships in the optimized tree $\mathcal{T}^\*$, which vary across nodes based on their community structure. The encoding tree construction automatically discovers semantically cohesive neighborhoods, making the neighbor count data-driven rather than manually specified.
>
> > Q2. Justify whether the greedy approach is sufficient to approximate a global optimum.
>
> Thanks for your question. The greedy approach is practically sufficient for several reasons.
>
> **Theoretically**, structural entropy optimization defines a clear objective function (Eq. 3-4), and our greedy strategy iteratively reduces entropy through BFS traversal with local operations. While global optimality is not guaranteed (graph partitioning is NP-hard), structural entropy's local minima have been proven effective for graph clustering in prior work.
>
> **Empirically**, the encoding tree demonstrates clear performance gains (ablation study), converges to stable structures, and exhibits hierarchical communities. The greedy approach achieves strong results with minimal cost (5% of training time), providing optimal efficiency-quality trade-off where global search is infeasible.
>
> > Q3. Discuss whether manual control could be introduced for the curriculum learning schedule.
>
> Thanks for your question. The current automatic approach offers significant advantages: $\lambda$ is computed via MMD (Eq. 7) without manual tuning, adapting dynamically based on data and training state, embodying a truly data-driven characteristic.
>
> However, manual control is possible for specific tasks. For instance, in scenarios with extreme modality imbalance or domain-specific requirements, one could introduce a manual adjustment factor to scale $\lambda$ or impose upper/lower bound constraints on its evolution. There exists a trade-off: manual control increases hyperparameter search burden and may sacrifice adaptivity, but could provide finer-grained control in specialized scenarios.
>
> We have revised the manuscript in Section 6 accordingly.
>
> > Q4. Discuss whether integrating pre-trained VLM knowledge would be a promising future direction.
>
> Thanks for your suggestion. Integrating pre-trained VLM knowledge is a highly promising direction with multiple potential approaches:
> - **Feature initialization:** using pre-trained VLM features as input for encoding tree construction could provide better semantic alignment;
> - **Knowledge distillation:** distilling VLM's cross-modal alignment knowledge into the hash model to enhance semantic understanding;
> - **Tree structure guidance:** leveraging VLM's attention maps or intermediate representations to guide Merge and Compress operations.
>
> However, challenges remain: maintaining computational efficiency of hash learning and adapting VLM to discrete hash spaces require careful design. We believe this represents a valuable future direction.
>
> We have revised the manuscript in Section 6 accordingly.

---

> ### Author Response · Authors · 2025-11-19
>
> We sincerely appreciate your thoughtful feedback and encouraging evaluation of our work. All the concerns and suggestions you raised have been carefully addressed in the revised manuscript with corresponding updates. We believe these revisions have significantly strengthened the paper, and we hope they meet your expectations. If you have any further questions or suggestions, please do not hesitate to let us know!

---

### Official Review · Reviewer_qurP · 2025-11-01

**Soundness:** 2
**Presentation:** 2
**Contribution:** 1
**Rating:** 2
**Confidence:** 4

**Summary:**

This paper proposes a novel unsupervised cross-modal retrieval framework named HINT (Hierarchical Encoding Tree with Modality Mixup). Specifically, HINT constructs a cross-modal encoding tree guided by hierarchical structural entropy, which organizes visual and textual representations into hierarchical communities. Each node in the encoding tree captures local semantic relations, while the overall tree structure preserves global semantic hierarchy. Based on this tree, the method synthesizes proxy samples for both modalities through a modality mixup strategy, enabling progressive alignment via curriculum learning. Experimental results show the proposed method can achieve good results.

**Strengths:**

1) The manuscript is well organized and easy to follow. The motivation, methodological design, and experimental setup are clearly presented, making the overall contribution understandable and coherent.

2) The proposed HINT framework achieves good and consistent results across multiple cross-modal retrieval benchmarks.

3) The appendix provides detailed additional analyses, including implementation details and supplementary experiments.

**Weaknesses:**

1) The contribution section uses overly strong and promotional language (e.g., “New Perspective,” “Coherent Framework,” “Outstanding Performance”), which is not fully justified by the presented methodology or experimental evidence. The proposed approach is conceptually sound, but the degree of novelty and improvement appears incremental rather than fundamentally transformative. The authors are encouraged to adopt a more objective tone and support such claims with stronger quantitative and qualitative evidence.

2)  The paper states that connecting the encoding tree with cross-modal hashing offers a new perspective. However, the idea of representing cross-modal relations in a hierarchical structure is not entirely new and has been discussed in previous studies (e.g., [ref1]) on hierarchical representation learning and cross-modal graph encoding. The proposed method mainly extends existing hierarchical modeling techniques rather than introducing a fundamentally different formulation or conceptual insight.

3) The paper only provides a brief time cost analysis in Table 4, 5 , without comparing the retrieval efficiency with existing cross-modal hashing methods (only 3). Moreover, as hashing-based models are typically valued for their efficiency, the absence of parameter-scale or computational complexity comparisons (e.g., model size, FLOPs, or training cost) weakens the empirical completeness of the work.

4) The comparative methods used in the experiments appear to be outdated, with most baselines coming from earlier studies (mainly up to 2023 or before). Recent advances in cross-modal hashing and retrieval from 2025 are not included.

5) The proposed modality mixup simply performs a linear interpolation between same-modality and cross-modality proxy samples. This operation lacks theoretical grounding on why such a linear combination can lead to better cross-modal alignment. Moreover, the parameter λ is only heuristically adjusted during training, and the method does not ensure semantic consistency when mixing features from heterogeneous modalities. As a result, the mixup process may blur modality-specific information rather than truly enhancing cross-modal representation learning.

6) While the paper presents a well-structured framework, its overall novelty appears limited. Most components—such as hierarchical encoding, proxy construction, and mixup-based alignment—are adaptations or recombinations of existing techniques.


[ref1]:  Jin W, Zhao Z, Zhang P, et al. Hierarchical cross-modal graph consistency learning for video-text retrieval[C]//Proceedings of the 44th International ACM SIGIR Conference on research and development in information retrieval. 2021: 1114-1124.

**Questions:**

See the weaknesses section.

---

> ### Author Response · Authors · 2025-11-19
> **Thanks for your review**
>
> Dear Reviewer qurP,
>
> Thank you for your thorough and constructive review. We genuinely appreciate your recognition of our well-organized presentation, consistent experimental results, and detailed analyses. Your insightful feedback has helped us significantly improve the manuscript. We have carefully addressed each of your concerns and revised the paper accordingly. Below, we provide detailed responses to each point.

---

> ### Author Response · Authors · 2025-11-19
>
> > W1. Adopt a more objective tone in the contribution section.
>
> Thanks for your comment. We have revised the contribution section to adopt a more objective tone. Specifically, we replaced *New Perspective* with *Hierarchical Modeling Approach*, *Coherent Framework* with *Integrated Framework,* and *Outstanding Performance* with *Competitive Performance*. We have revised the manuscript in Section 1 accordingly.
>
>
> > W2. Clarify the novelty compared to hierarchical methods like HCGC.
>
> Thanks for your comment. We acknowledge that hierarchical structures are an important research direction in cross-modal retrieval. However, HINT differs fundamentally from existing works [1,2,3,4] in the following dimensions:
>
> - **Task Difference.** Methods [1,2,4] focus on supervised video-text retrieval, learning continuous embedding spaces. HINT addresses unsupervised cross-modal hashing. This task difference leads to fundamentally different challenges: how to discover hierarchical semantic structures from sparse pairwise relationships in an unsupervised manner.
>
> - **Construction Difference.** The hierarchical structures in [1,4] are predefined, based on rules and external tools (e.g., semantic parsers, object detectors) or preset granularities. Method [2] constructs query-specific syntactic trees. Method [3] adopts architectural hierarchies rather than data structure hierarchies. In contrast, HINT's encoding tree is data-driven, automatically discovered via structural entropy optimization, representing the dataset's intrinsic structure.
>
> - **Utilization Difference.** Methods [1,4] perform matching or alignment between predefined fixed levels. Method [3] achieves progressive dimensionality reduction through network architecture. HINT introduces a novel mechanism: synthesizing proxy samples based on the learned encoding tree and achieving progressive alignment through curriculum-based modality mixup. We leverage the hierarchical tree to guide an adaptive alignment process, rather than directly matching at fixed levels.
>
> We have revised Section 3 accordingly.
>
> [1] Hierarchical cross-modal graph consistency learning for video-text retrieval, SIGIR 2021.
>
> [2] Tree-Augmented Cross-Modal Encoding for Complex-Query Video Retrieval, SIGIR 2020.
>
> [3] Hierarchical Consensus Hashing for Cross-Modal Retrieval, IEEE TMM 2023.
>
> [4] HTVR: Hierarchical text-to-video retrieval based on relative similarity, PR 2025.
>
>
> > W3. Provide comprehensive efficiency analysis including model complexity and training cost.
>
> Thanks for your comment. We have now supplemented comprehensive comparisons of model complexity and training efficiency.
>
> **Retrieval Efficiency:** HINT and all hashing methods use the same retrieval mechanism (Hamming distance computation), thus having identical retrieval efficiency. Table 4 shows hash-based retrieval achieves 26-31× speedup over dense vector approaches across different code lengths.
>
> **Training Efficiency:** We have added a comprehensive comparison in Table 5 on MIRFlickr-25K with 128-bit codes:
>
> | Method | Params (M) | Total Time (min) | MAP (%) |
> |--------|------------|------------------|---------|
> | UCHSTM | 52.5 | 182 | 72.3 |
> | UCCH | 88.0 | 58 | 73.2 |
> | UDDH | 28.9 | 45 | 74.6 |
> | HuggingHash+ | 40.1 | 211 | 74.5 |
> | DEMO | 86.2 | 59 | 74.3 |
> | HINT | 84.8 | 61 | 75.5 |
>
> HINT has 84.8M parameters and requires 61 minutes total training time, both comparable to DEMO and UCCH. The hierarchical encoding tree construction takes only <5% overhead. Despite similar training costs, HINT achieves 1.2-2.3% MAP improvement over baselines, demonstrating an excellent efficiency-performance trade-off.
>
> We have revised Section 5 and updated the Appendix accordingly.

---

> ### Author Response · Authors · 2025-11-19
>
> > W4. Include recent methods for comparison.
>
> Thanks for your comment. Actually, we note that our experiments already include three recent 2024 methods ( UDDH, HuggingHash+, DEMO ) that represent the state-of-the-art in unsupervised cross-modal hashing. To further strengthen our comparison, we have now included two recent 2025 works: GCRH [1] and VTM-UCH [2].
>
> GCRH proposes a contrastive-and-reconstructive learning framework that integrates global graph contrastive learning and local graph reconstruction to reduce the modality gap. VTM-UCH introduces vision-guided text mining with community similarity quantization to address text modality deficiency and improve hash code distribution.
>
> As shown in the updated Table 1, HINT consistently outperforms both methods across all datasets and code lengths. The additional comparison is shown below:
>
> **Image → Text:**
>
> | Method | MIRFlickr-25K | | | | NUS-WIDE | | | | MS-COCO | | | |
> |--------|------|------|------|------|------|------|------|------|------|------|------|------|
> | | 16 | 32 | 64 | 128 | 16 | 32 | 64 | 128 | 16 | 32 | 64 | 128 |
> | DEMO | 71.8 | 73.3 | 73.4 | 74.3 | 64.6 | 64.8 | 66.2 | 66.4 | 57.5 | 57.8 | 58.6 | 60.5 |
> | GCRH | 71.0 | 72.2 | 72.7 | 73.3 | 63.9 | 64.0 | 65.3 | 65.5 | 56.8 | 57.0 | 57.8 | 59.6 |
> | VTM-UCH | 71.9 | 73.6 | 73.9 | 74.5 | 64.6 | 65.1 | 66.0 | 66.6 | 57.6 | 58.2 | 58.8 | 60.3 |
> | **HINT** | **72.9** | **74.4** | **75.1** | **75.5** | **65.1** | **65.5** | **66.5** | **67.3** | **58.5** | **59.5** | **60.4** | **61.1** |
>
> **Text → Image:**
>
> | Method | MIRFlickr-25K | | | | NUS-WIDE | | | | MS-COCO | | | |
> |--------|------|------|------|------|------|------|------|------|------|------|------|------|
> | | 16 | 32 | 64 | 128 | 16 | 32 | 64 | 128 | 16 | 32 | 64 | 128 |
> | DEMO | 70.8 | 71.9 | 72.2 | 72.8 | 65.4 | 65.5 | 66.9 | 67.1 | 57.2 | 57.9 | 58.3 | 59.7 |
> | GCRH | 70.1 | 71.0 | 71.3 | 71.8 | 64.5 | 64.7 | 66.0 | 66.2 | 56.5 | 57.3 | 57.4 | 58.8 |
> | VTM-UCH | 71.4 | 72.1 | 72.4 | 73.0 | 65.6 | 65.7 | 66.9 | 67.3 | 57.4 | 58.0 | 58.3 | 59.6 |
> | **HINT** | **72.0** | **73.1** | **74.0** | **74.6** | **66.0** | **66.6** | **67.3** | **67.8** | **58.2** | **59.0** | **59.8** | **60.8** |
>
> These results demonstrate that HINT consistently achieves superior performance compared to the latest methods.
>
> We have revised the manuscript in Section 4 and updated Table 1 accordingly.
>
> [1] Graph Contrastive-and-Reconstructive Hashing for Unsupervised Cross-Modal Retrieval. 2025
>
> [2] Vision-guided Text Mining for Unsupervised Cross-modal Hashing with Community Similarity Quantization. AAAI 2025.
>
>
> > W5. Clarify the theoretical foundation and adaptive nature of modality mixup.
>
> Thanks for your comment. We respectfully clarify that several critiques may be based on misunderstandings of the paper.
>
> - **Regarding $\lambda$ being "heuristically adjusted":** This may not correct. Eq. 7 explicitly defines $\lambda$ as $\lambda = \widehat{MMD}(\rho(m^{same}, \mathcal{B}), \rho(m^{cross}, \mathcal{B}))$, where MMD is a data-driven metric quantifying the modality gap. Figure 5 demonstrates $\lambda$ evolving from ~0.8 to <0.5, confirming its adaptive nature rather than being a fixed hyperparameter.
>
> - **Regarding "simple linear interpolation" and theoretical grounding:** The MMD-controlled dynamic interpolation implements curriculum learning [2] for progressive alignment: initially, large $\lambda$ makes $b_i^{mix}$ rely on same-modal features (easier task); as $\lambda$ decreases, cross-modal features become prominent (harder task). The theoretical foundation is manifold regularization [1]: feature-space interpolation between proxy samples forces both modalities to find a shared semantic space where their interpolation remains meaningful for $\mathcal{L}_{hash}$.
>
> - **Regarding "blurring modality-specific information":** HINT mixes semantically purified proxy samples (Eq. 6) constructed via tree-based neighborhood aggregation, not raw heterogeneous features. This ensures semantic coherence before mixing, similar to advanced cross-modal mixup strategies [3] that avoid manifold intrusion.
>
> - **Regarding "not ensuring semantic consistency":** This may not correct. Section 3.4 explicitly introduces L_{con} (Eq. 11) using KL divergence to enforce semantic consistency. HINT employs dual guarantees: (1) structural-tree-based proxy construction for local consistency, and (2) loss-based-$\mathcal{L}_{con}$ for global distributional consistency.
>
>
> [1] Manifold mixup: Better representations by interpolating hidden states. ICML 2019.
>
> [2] ProCLIP: Progressive Vision-Language Alignment via LLM-based Embedder. arXiv 2025.
>
> [3] Modality-adaptive mixup and invariant decomposition for RGB-infrared person re-identification. AAAI 2022.

---

> ### Author Response · Authors · 2025-11-19
>
> > W6. Clarify the system-level innovation beyond individual components.
>
> Thanks for your comment. HINT's novelty lies in its unified goal-driven design where all components synergistically address progressive sub-challenges toward learning discriminative cross-modal hash codes under sparse supervision.
>
> - **Unified Goal-Driven Design.** All components serve a unified objective: learning discriminative cross-modal hash codes without annotations. They form a progressive solution to hierarchical sub-challenges: (1) the encoding tree addresses sparse supervision by discovering hierarchical semantic structures; (2) proxy samples bridge the heterogeneous modality gap as semantic intermediaries; (3) adaptive mixup achieves progressive alignment through curriculum learning; (4) consistency learning enforces global distributional constraints. Each component's output feeds the next, forming an organic pipeline where removing any element breaks the entire solution.
>
> - **Empirical Validation.** Our ablation study confirms this integration. The performance improves from 73.2 MAP (V1 baseline) to 75.5 MAP (full model), with cumulative 2.3 MAP gain. This demonstrates that the whole exceeds the sum of its parts. Each component's contribution amplifies through system-level integration.
>
> - **Methodological Contribution.** For unsupervised cross-modal hashing, HINT proposes a hierarchical-first approach: discovering semantic structures before alignment, rather than directly optimizing cross-modal similarity as in existing flat alignment methods. This structure-then-align strategy enables more effective learning under sparse pairwise supervision.
>
> We have revised the manuscript in Sections 1 and 3 accordingly.

---

> ### Author Response · Authors · 2025-11-19
>
> We are grateful for your constructive feedback, which has significantly improved the quality of our manuscript. We have worked diligently to address all your concerns in the revised version and hope the changes meet your expectations. If you have any further questions or need additional clarifications, please do not hesitate to let us know. Thank you again for your valuable time and insights.

---

### Author Response · Authors · 2025-11-19
**Revision Summary**

We sincerely thank the Area Chair and Reviewers for their constructive feedback. In this revision, we have systematically addressed all concerns. The major updates are summarized below.

**1. Strengthened Experiments**
To address concerns regarding baselines and efficiency (Reviewers qurP, LKH2):
*   **New SOTA Comparisons:** We integrated two latest 2025 methods (GCRH and VTM-UCH). HINT consistently outperforms them across all datasets and settings.
*   **Efficiency Analysis:** We provided a detailed analysis of model complexity, training time, and retrieval speed, demonstrating HINT's superior efficiency-performance trade-off.
*   **Performance Asymmetry:** We clarified that the significant gains in Text$\rightarrow$Image from our hierarchical structure, which effectively mitigate the sparsity and lower quality of text features.

**2. Methodological Clarifications**
To address questions on design rationale and mechanisms (Reviewers kUAR, LKH2, Y8zT):
*   **System Synergy:** We articulated that the encoding tree, proxy samples, and mixup mechanism form a synergistic dependency chain, essential for the framework's end-to-end effectiveness.
*   **Static Tree & Greedy Strategy:** We justified the static tree design for its training stability and computational efficiency. We further explained that our greedy structural entropy optimization offers a robust and practical solution for the NP-hard community discovery problem.
*   **Proxy Construction:** We distinguished our **sample-level proxies** from class-level prototypes and justified the use of simple averaging, as the encoding tree inherently ensures the semantic purity of neighbors.
*   **Curriculum Mixup:** We clarified that the mixup parameter $\lambda$ is data-driven via MMD (not manually tuned), implementing an adaptive curriculum that progressively bridges the modality gap.

**3. Writing & Future Outlook**
To improve clarity and expand scope (Reviewers qurP, LKH2, kUAR, Y8zT):
*   **Contributions & Related Work:** We rewrote these sections to explicitly differentiate HINT from existing hierarchical methods, focusing on our unique unsupervised structure discovery.
*   **Expanded Future Directions:** We significantly broadened the discussion on future work, outlining six promising directions including weighted aggregation, streaming data adaptation, semi-supervised extensions, and VLM integration.


We are grateful for the opportunity to improve our paper with this feedback.  For our detailed/point-by-point responses to each reviewer, please see the respective threads.

---

### Meta-Review · Area_Chair_JjAV · 2025-12-29

**Summary:**

The main concerns for this paper come from the experimental comparisons, model complexity, training time, retrieval speed, performance asymmetry, methodological clarifications, and others.

**Reviewer Concerns:**

Added more recently published work in 2025 showing good performance across all datasets and settings.
The authors articulated that the encoding tree, proxy samples, and mixup mechanism form a synergistic dependency chain.
They clarified the static tree & greedy strategy, proxy construction, curriculum mixup, and other related points.

**Reviewer Scores:**

All the reviewers held positive points about this paper before the incident, and the final scores are 8, 8, 6, 6.

---

### Decision · Program_Chairs · 2026-01-26

Accept (Poster)